# CondTSF: One-line Plugin of Dataset Condensation for Time Series Forecasting

**Jianrong Ding[1,2]\*, Zhanyu Liu[1]\*, Guanjie Zheng[1]†, Haiming Jin[1], Linghe Kong[1]**

[1] School of Electronic Information and Electrical Engineering, Shanghai Jiao Tong University
[2] Zhiyuan College, Shanghai Jiao Tong University
`{rafaelding,zhyliu00,gjzheng,jinhaiming,linghe.kong}@sjtu.edu.cn`

## Abstract

*Dataset condensation* is a newborn technique that generates a small dataset that can be used in training deep neural networks (DNNs) to lower storage and training costs. The objective of dataset condensation is to ensure that the model trained with the synthetic dataset can perform comparably to the model trained with full datasets. However, existing methods predominantly concentrate on classification tasks, posing challenges in their adaptation to time series forecasting (TS-forecasting). This challenge arises from disparities in the evaluation of synthetic data. In classification, the synthetic data is considered well-distilled if the model trained with the full dataset and the model trained with the synthetic dataset yield identical labels for the same input, regardless of variations in output logits distribution. Conversely, in TS-forecasting, the effectiveness of synthetic data distillation is determined by the distance between predictions of the two models. The synthetic data is deemed well-distilled only when all data points within the predictions are similar. Consequently, TS-forecasting has a more rigorous evaluation methodology compared to classification. To mitigate this gap, we theoretically analyze the optimization objective of dataset condensation for TS-forecasting and propose a new one-line plugin of dataset condensation for TS-forecasting designated as Dataset **Cond**ensation for **T**ime **S**eries **F**orecasting (CondTSF) based on our analysis. Plugging CondTSF into previous dataset condensation methods facilitates a reduction in the distance between the predictions of the model trained with the full dataset and the model trained with the synthetic dataset, thereby enhancing performance. We conduct extensive experiments on eight commonly used time series datasets. CondTSF consistently improves the performance of all previous dataset condensation methods across all datasets, particularly at low condensing ratios.

## 1 Introduction

Dataset condensation is a strategy for mitigating the computational demands of training large models on extensive datasets. It is pointed out by previous works[15, 32] that building foundation models[14, 10, 6, 35, 2, 44] on time series forecasting (TS-forecasting) have become a hot topic. However, fine-tuning these large models using full time series datasets can entail considerable computational overhead. Hence, the employment of dataset condensation techniques becomes imperative. In recent years, various methods have been proposed in the field of dataset condensation, such as matching-based methods[51, 49, 3, 20, 38, 7, 5, 41, 50, 52, 36] and kernel methods[33, 55]. To date, dataset condensation methods have achieved success in classification tasks, including image classification[8, 11, 22], graph classification[17, 16, 23, 43, 25, 9, 27] and time series classification[26].

---

\* Co-primary author.    † Corresponding author.

38th Conference on Neural Information Processing Systems (NeurIPS 2024).

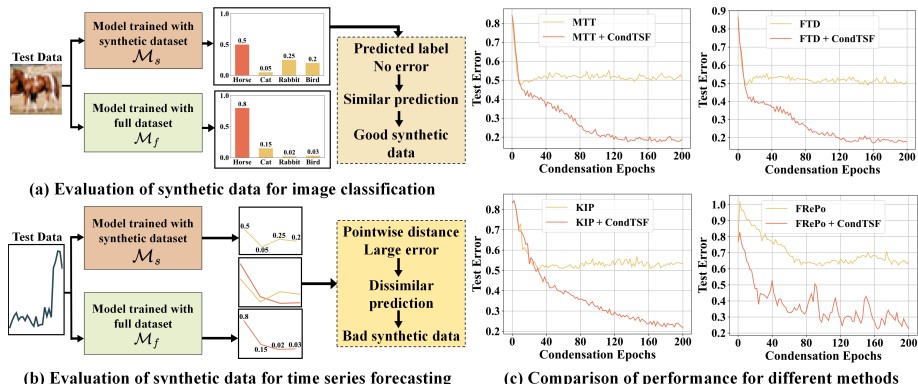

Figure 1: **Left:** Difference between evaluation of dataset condensation for classification tasks and time series forecasting tasks. **Right:** Comparison in performance of previous methods with and without CondTSF.

However, directly applying these dataset condensation methods designed for classification to the domain of time series forecasting (TS-forecasting) results in performance degradation. The objective of dataset condensation is to generate a synthetic dataset so that when the model $\mathcal{M}_s$ trained with the synthetic dataset and the model $\mathcal{M}_f$ trained with the full dataset are given identical input, the two models output **similar predictions**. However, the concept of **similar prediction** differs between classification and TS-forecasting. In classification, as shown in Fig.1(a), predictions are considered similar if $\mathcal{M}_s$ and $\mathcal{M}_f$ assign the same class label, irrespective of differences in the distribution of output logits. Conversely, in TS-forecasting, as illustrated in Fig.1(b), the similarity of predictions from $\mathcal{M}_s$ and $\mathcal{M}_f$ is indicated by the mean squared distance of the predictions. The predictions are deemed similar only when all data points within the predictions are similar. This distinction in evaluation indicates TS-forecasting imposes more stringent criteria in discerning **similar predictions** compared to classification. It poses a challenge that previous dataset condensation methods based on classification fail to provide adequate assurance for the similarity between predictions of $\mathcal{M}_s$ and $\mathcal{M}_f$ within the realm of TS-forecasting.

To mitigate the gap, we propose a novel one-line dataset condensation plugin designed specifically for TS-forecasting called **Cond**ensation for **T**ime **S**eries **F**orecasting (CondTSF) based on our theoretical analysis. We first formulate the optimization objective of dataset condensation for TS-forecasting. Then we transform the original optimization objective into minimizing the distance between predictions of $\mathcal{M}_s$ and $\mathcal{M}_f$. Furthermore, to minimize the distance between predictions of $\mathcal{M}_s$ and $\mathcal{M}_f$, we decompose the task into minimizing two terms, namely **gradient term** and **value term**. We theoretically prove that plugging CondTSF into previous methods can minimize the **value term** and **gradient term** synchronously. Therefore, CondTSF serves as an effective plugin to boost the performance of dataset condensation for TS-forecasting. As depicted in Fig.1(c), plugging CondTSF into previous methods yields a significant enhancement in performance.

In short, our contributions can be summarized as follows.

- To the best of our knowledge, we are the first to explore dataset condensation for TS-forecasting. We conduct a theoretical analysis of the optimization objective of dataset condensation for TS-forecasting, breaking it down into two optimizable terms to facilitate improved optimization.

- Leveraging insights from our theoretical analysis of TS-forecasting, we propose a simple yet effective dataset condensation plugin CondTSF. Plugging CondTSF into existing methods enables synchronous optimization of the two terms, leading to performance enhancement.

- We conduct extensive experiments on eight widely used time series datasets to prove the effectiveness of CondTSF. CondTSF notably improves the performance of all previous dataset condensation methods across all datasets, particularly under low condensing ratios.

## 2 Related Works

**Time Series Forecasting:** Time series forecasting (TS-forecating) is the task of using historical, time-stamped data to predict future values. Previous works utilize different methods to achieve better

performance. These models can be mainly categorized into 3 types. **(1)** Transformer-based Models: Transformer[40] have shown great success in natural language processing, and models based on transformers[53, 42, 24, 54] emerged in TS-forecasting fields. **(2)** MLP-based Models: Efforts to use MLP-based models have been put into TS-forecasting in recent years[47] since DLinear[45] triumph transformer-based models with a simple MLP structure. **(3)** Patch-based Models: These models[34, 48, 28, 29] focused on learning representation cross patches instead of learning attention at each time point. Therefore they used a patching strategy before feeding the data to transfomers.

**Dataset Condensation:** Dataset condensation is a task that aims at distilling a large dataset into a smaller one so that when a model is trained on the small synthetic dataset and the full dataset separately, the testing performances of the trained models are similar. Previous works related to dataset condensation can be divided into 3 classes below. **(1)** Coreset Selecting Methods: These methods aim at selecting data with representative features from source dataset to construct a synthetic dataset[1, 4, 12, 37, 39]. **(2)** Matching-based Methods: These methods aim at minimizing a specific metric surrogate model learned from source dataset and synthetic dataset. The defined metrics are different, including gradient[51, 18, 46], features from the same class[41], distribution of synthetic data[50, 52] and training trajectories[3, 5, 7, 11, 8]. **(3)** Kernel-based Methods: These methods aim at obtaining a closed-form solution for the optimization problem utilizing kernel ridge-regression[20, 33]. In this way, the bi-level optimization problem of dataset condensation is reduced to a single-level optimization problem. Based on these results, the following works have made significant progress in different areas, including decreasing training cost and time[55], improving performance[30, 31].

## 3 Preliminaries

**Dataset Condensation for TS-forecasting** Given a time series dataset, we split the dataset into a train set and a test set. In this paper, we denote the train set as $f$ and the test set as $x$. We denote the synthetic dataset as $s$. The synthetic dataset $s$ is a small dataset distilled from the full train set $f$. Train set $f$, test set $x$, and synthetic dataset $s$ are all vectors. We employ $\mathcal{M}_\theta$ as a neural network parameterized by $\theta$. Without losing generality, we suppose the model $\mathcal{M}_\theta$ is using historical sequence $x_{t:t+m}$ with length $m$ to predict future sequence $x_{t+m:t+m+n}$ with length $n$. Given the test set $x$, we formulate the test error of $\mathcal{M}_\theta$ as the error between the prediction of $\mathcal{M}_\theta$ on test input $x_{t:t+m}$ and the test label $x_{t+m:t+m+n}$, as shown in Eq.1.

$$\mathcal{L}_{test}(\mathcal{M}_\theta, x) \triangleq \sum_t ||\mathcal{M}_\theta(x_{t:t+m}) - x_{t+m:t+m+n}||^2 \tag{1}$$

During **dataset condensation process**, a distribution of initial model parameters $P_\theta$ is available for training model parameter sampling, and the full train set $f$ is available for condensation. Subsequently, a synthetic dataset $s$ is distilled from the full train set $f$ using dataset condensation methods. During **testing process**, initial testing model parameter $\theta_{0,test}$ is sampled from $P_\theta$. Since $\theta_{0,test}$ is sampled in the testing process, it's unavailable during the previous dataset condensation process. Then model parameters $\theta_{s,test}$ and $\theta_{f,test}$ are obtained by training initial testing parameter $\theta_{0,test}$ on synthetic dataset $s$ and the full train set $f$ respectively. The objective of dataset condensation is to ensure model $\mathcal{M}_{\theta_{s,test}}$ and $\mathcal{M}_{\theta_{f,test}}$ have comparable performance on test set $x$. Therefore the practical optimization objective is to ensure that model $\mathcal{M}_{\theta_{s,test}}$ trained with synthetic dataset $s$ minimizes the test error $\mathcal{L}_{test}$ on test set $x$. The optimization objective is formulated as Eq.2.

$$\min_s \mathcal{L}_{test}(\mathcal{M}_{\theta_{s,test}}, x) \tag{2}$$

## 4 Method

Since test set $x$ is not available during the dataset condensation process, the original optimization objective for dataset condensation in Eq.2 is non-optimizable. To mitigate this gap, in the following sections, we transform the non-optimizable objective into two distinct optimizable terms. Then we develop methods to optimize the two terms, thereby indirectly optimizing the original objective.

### 4.1 Decomposition

In this section, we decompose the optimization objective of dataset condensation in Eq.2 into two optimizable terms for better optimization. In the testing process, the initial testing model parameter

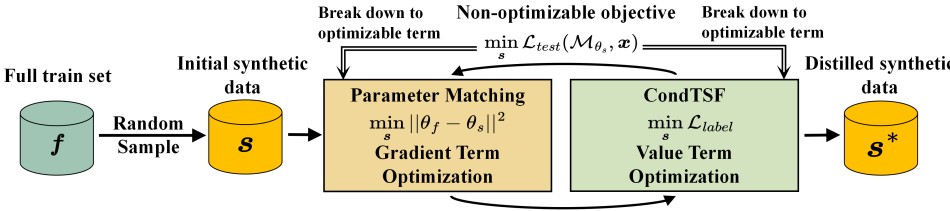

Figure 2: Complete process of dataset condensation using CondTSF.

$\theta_{0,test}$ is sampled from a distribution of initial model parameters $P_\theta$. Then we train $\theta_{0,test}$ on the synthetic dataset $s$ to get model parameter $\theta_{s,test}$, and train $\theta_{0,test}$ on the full train set $f$ to get model parameter $\theta_{f,test}$. Given test dataset $x$, the optimization objective is formulated as Eq.3.

$$\min_s \mathcal{L}_{test}(\mathcal{M}_{\theta_{s,test}}, x)$$

$$\text{where } \mathcal{L}_{test}(\mathcal{M}_{\theta_{s,test}}, x) = \sum_t ||\mathcal{M}_{\theta_{s,test}}(x_{t:t+m}) - x_{t+m:t+m+n}||^2 \tag{3}$$

Meanwhile, there is a non-optimizable error $\epsilon$ between the prediction of model $\mathcal{M}_{\theta_{f,test}}$ and the true label from the test dataset, which is formulated in Eq.4.

$$x_{t+m:t+m+n} = \mathcal{M}_{\theta_{f,test}}(x_{t:t+m}) + \epsilon \tag{4}$$

Then we decompose the upper bound of $\mathcal{L}_{test}(\mathcal{M}_{\theta_{s,test}}, x)$ into two terms, as shown in Thm.1. We utilize Taylor Expansion in the proof of Thm.1. For each real test data $x_{t:t+m}$, we can arbitrarily choose position $t'$ and get synthetic data $s_{t':t'+m}$. Then we can perform Taylor Expansion with $s_{t':t'+m}$ to obtain the value of $\mathcal{M}_{\theta_{s,test}}(x_{t:t+m})$ and $\mathcal{M}_{\theta_{f,test}}(x_{t:t+m})$.

**Theorem 1.** *Given arbitrary synthetic data $s_{t':t'+m}$, the upper bound of the optimization objective of dataset condensation $\mathcal{L}_{test}(\mathcal{M}_{\theta_{s,test}}, x)$ can be formulated as such*

$$\mathcal{L}_{test}(\mathcal{M}_{\theta_{s,test}}, x) \leq \sum_t ||\epsilon||^2 + \underbrace{||\mathcal{M}_{\theta_{s,test}}(s_{t':t'+m}) - \mathcal{M}_{\theta_{f,test}}(s_{t':t'+m})||^2}_{\textit{Value Term}}$$

$$+ \underbrace{||(\nabla\mathcal{M}_{\theta_{s,test}}(s_{t':t'+m}) - \nabla\mathcal{M}_{\theta_{f,test}}(s_{t':t'+m}))^\top (x_{t:t+m} - s_{t':t'+m})||^2}_{\textit{Gradient Term}} \tag{5}$$

To prove Thm.1, we use linear models for further analysis since linear models can be both effective and efficient in TS-forecasting[45]. Given a linear model $\mathcal{M}_\theta(x) = \theta x$, its second and higher order gradient is zero. Therefore first-order Taylor Expansion is sufficient to obtain the accurate prediction of the model. Meanwhile, if $\mathcal{M}_\theta$ is a non-linear model, we ignore the higher-order terms of Taylor Expansion. We prove Thm.1 by applying the property of the first-order Taylor Expansion and triangular inequality of norm functions. The complete proof is in App.A.1. Hence we decompose the optimization objective of dataset condensation for TS-forecasting into two optimizable terms, namely **value term** and **gradient term**. For **value term**, it ensures $\mathcal{M}_{\theta_{s,test}}$ and $\mathcal{M}_{\theta_{f,test}}$ are similar in prediction values. For **gradient term**, it ensures the predictions of $\mathcal{M}_{\theta_{s,test}}$ and $\mathcal{M}_{\theta_{f,test}}$ are similar in gradient. Optimizing these two terms can optimize the upper bound of the original optimization objective, and therefore indirectly optimize the original optimization objective in Eq.3.

### 4.2 Gradient Term Optimization

We develop a method to optimize **gradient term** in this section. Given a linear model $\mathcal{M}_\theta(x) = \theta x$, its gradient on input is $\nabla\mathcal{M}_\theta(x) = \theta^\top$. It indicates that the gradient of a linear model on input is the parameter of the model. We apply Cauchy-Schwarz Inequality to the gradient term and get its upper bound. We reformulate the gradient term and get its upper bound as shown in Eq.6.

$$||(\nabla\mathcal{M}_{\theta_{s,test}}(s_{t':t'+m}) - \nabla\mathcal{M}_{\theta_{f,test}}(s_{t':t'+m}))^\top (x_{t:t+m} - s_{t':t'+m})||^2 \textit{ (Gradient Term)}$$

$$= ||(\theta_{s,test} - \theta_{f,test})(x_{t:t+m} - s_{t':t'+m})||^2 \tag{6}$$

$$\leq ||\theta_{s,test} - \theta_{f,test}||^2 \cdot ||x_{t:t+m} - s_{t':t'+m}||^2$$

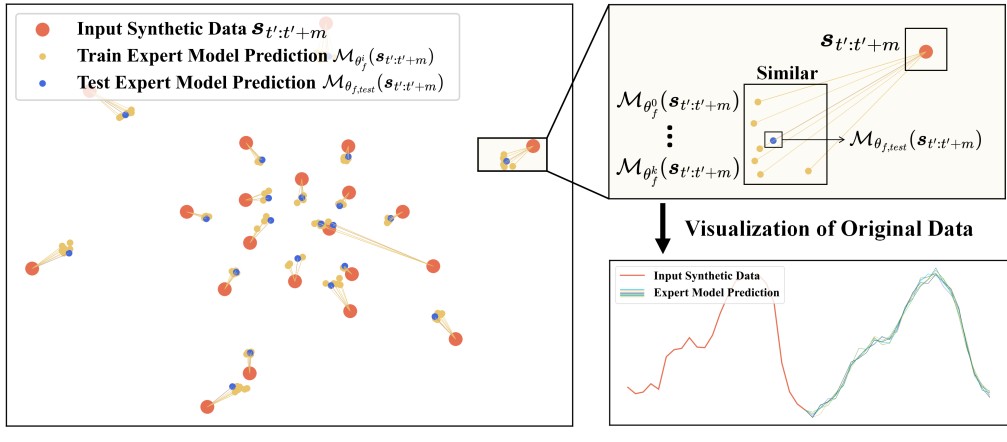

Empirically illustrate: $\mathcal{M}_{\theta_{f,test}}(\boldsymbol{s}_{t':t'+m}) \approx \mathcal{M}_{\theta_f^0}(\boldsymbol{s}_{t':t'+m}) \approx \mathcal{M}_{\theta_f^1}(\boldsymbol{s}_{t':t'+m}) \approx \cdots \approx \mathcal{M}_{\theta_f^k}(\boldsymbol{s}_{t':t'+m})$

Figure 3: Given the same synthetic data as input, all expert models trained on the full train set $\boldsymbol{f}$ provide similar predictions. The initial parameters of the models are sampled from the same distribution $P_\theta$. The visualization of this figure utilized MDS[19] algorithm for dimension reduction.

Since test data $\boldsymbol{x}_{t:t+m}$ is not available during the dataset condensation process, the distance between synthetic data and test data $||\boldsymbol{x}_{t:t+m} - \boldsymbol{s}_{t':t'+m}||^2$ is not optimizable. Therefore we only need to optimize the distance between parameters $||\theta_{s,test} - \theta_{f,test}||^2$. All previous dataset condensation methods based on parameter matching can minimize this distance. Here we utilize MTT[3] as an example to clarify the optimization process. The optimization objective of trajectory matching is

$$\min_{\boldsymbol{s}} \frac{||\theta_{f,test} - \theta_{s,test}||^2}{||\theta_{f,test} - \theta_{0,test}||^2} \tag{7}$$

However, since $\theta_{s,test}$ and $\theta_{f,test}$ are trained from testing initial parameter $\theta_{0,test} \sim P_\theta$, they are not available during dataset condensation process. Therefore, in practice, we sample $\theta_0^0, \ldots, \theta_0^k \sim P_\theta$ as initial parameters during dataset condensation process. The initial parameters are trained on synthetic dataset $\boldsymbol{s}$ and full train set $\boldsymbol{f}$ respectively to get $\theta_s^0, \ldots, \theta_s^k$ and $\theta_f^0, \ldots, \theta_f^k$. Then we substitute $\theta_{s,test}$, $\theta_{f,test}$ and $\theta_{0,test}$ in Eq.7 with parameters sampled in dataset condensation, making the optimization objective optimizable. The practical optimization objective is shown in Eq.8.

$$\min_{\boldsymbol{s}} \sum_{i=0}^{k} \frac{||\theta_f^i - \theta_s^i||^2}{||\theta_f^i - \theta_0^i||^2} \tag{8}$$

In practice, $\theta_0^0, \ldots, \theta_0^k$ and $\theta_f^0, \ldots, \theta_f^k$ are sampled, trained, and stored in a parameter buffer before dataset condensation process. It can be concluded that using trajectory matching methods is intuitively minimizing the distance between $\theta_s^i$ and $\theta_f^i$ for all initial parameters $\theta_0^i \sim P_\theta$. By minimizing the upper bound of the gradient term, trajectory matching methods indirectly optimize the gradient term.

## 4.3 Value Term Optimization

We develop an optimization objective to optimize the **value term** in this section. Since $\theta_{f,test}$ is trained from $\theta_{0,test}$, it's unavailable in dataset condensation process. To mitigate this gap, we prove that although $\theta_{f,test}$ is unavailable in dataset condensation process, its prediction $\mathcal{M}_{\theta_{f,test}}(\boldsymbol{s}_{t':t'+m})$ is still available. To prove this statement, we sample initial model parameters $\theta_0^0, \ldots, \theta_0^k$ from $P_\theta$. Then $\theta_0^0, \ldots, \theta_0^k$ are all trained with the same full train set $\boldsymbol{f}$. After training, we get parameters $\theta_f^0, \ldots, \theta_f^k$. It is observed that models $\mathcal{M}_{\theta_f^0}, \ldots, \mathcal{M}_{\theta_f^k}$ predict similarly given arbitrary synthetic data $\boldsymbol{s}_{t':t'+m}$ as input.

Since initial testing parameter $\theta_{0,test}$ is also sampled from the same distribution $P_\theta$ and $\theta_{f,test}$ is trained from $\theta_{0,test}$ using the same full train set $\boldsymbol{f}$, the prediction of $\mathcal{M}_{\theta_{f,test}}$ is similar to predictions of an arbitrary expert model $\mathcal{M}_{\theta_f^i}$. The conclusion is formulated in Eq.9.

$$\mathcal{M}_{\theta_f,test}(\boldsymbol{s}_{t':t'+m}) \approx \mathcal{M}_{\theta_f^0}(\boldsymbol{s}_{t':t'+m}) \approx \mathcal{M}_{\theta_f^1}(\boldsymbol{s}_{t':t'+m}) \approx \cdots \approx \mathcal{M}_{\theta_f^k}(\boldsymbol{s}_{t':t'+m}) \tag{9}$$

**Algorithm 1** Dataset Condensation with CondTSF (MTT[3] as backbone)

---

**Input:** Synthetic data $s$; Parameter buffer $\{(\theta_0, \theta_f)\}^k$; Synthetic learning rate $\alpha$; Trajectory matching epochs $N$; Total condensation epochs $E$; Additive update ratio $\beta$; Gap of epochs $G$ between using CondTSF

**Output:** Optimized synthetic data $s$

1: Split $s$ into training sets $\{(s_{t:t+m}, s_{t+m:t+m+n})\}^l$
2: **for** each condensation epoch $e$ in range $E$ **do**
3:     **if** $e \mod G \neq 0$ **then**
4:         Sample $(\theta_0^i, \theta_f^i)$ from $\{(\theta_0, \theta_f)\}^k$
5:         Initialize student parameter $\hat{\theta}_0 \leftarrow \theta_0^i$
6:         **for** each trajectory matching epoch $j$ in range $N$ **do**    */\*MTT[3] trajectory matching\*/*
7:             Train $\mathcal{M}_{\hat{\theta}_j}$ with synthetic data
8:             $\hat{\theta}_{j+1} \leftarrow \hat{\theta}_j - \alpha\nabla\mathcal{L}(\mathcal{M}_{\hat{\theta}_j}(s_{t:t+m}), s_{t+m:t+m+n})$ for all synthetic data
9:         $\mathcal{L}_{param} \leftarrow ||\theta_f^i - \hat{\theta}_N||^2 / ||\theta_f^i - \theta_0^i||^2$
10:        Update synthetic data $s$ with respect to $\mathcal{L}_{param}$    */\*Optimize gradient term\*/*
11:     **else**
12:         **for** each train sample $(s_{t:t+m}, s_{t+m:t+m+n})$ in training sets **do**
13:             Choose an arbitrary expert model with parameter $\theta_f^i$ from $\{(\theta_0, \theta_f)\}^k$
14:             $y \leftarrow \mathcal{M}_{\theta_f^i}(s_{t:t+m})$
15:             $s_{t+m:t+m+n} \leftarrow (1-\beta) \cdot s_{t+m:t+m+n} + \beta \cdot y$   */\*Optimize value term\*/*
16: **return** $s$

---

Experiments have proved Eq.9 in Fig.3. As shown in Fig.3, for each synthetic data input $s_{t':t'+m}$ (orange points), the predictions of corresponding expert models (yellow and blue points) are similar. Therefore, although $\theta_{f,test}$ is unavailable in the dataset condensation process, its prediction $\mathcal{M}_{\theta_{f,test}}(s_{t':t'+m})$ can be obtained using the prediction of an arbitrary expert model $\mathcal{M}_{\theta_f^i}(s_{t':t'+m})$. Now we reformulate the value term and transform it into a practical optimization objective. Firstly, We formulate the upper bound of the value term as shown in Thm.2.

**Theorem 2.** *The upper bound of the value term can be formulated as such*

$$||\mathcal{M}_{\theta_{s,test}}(s_{t':t'+m}) - \mathcal{M}_{\theta_{f,test}}(s_{t':t'+m})||^2 \leq 2 \cdot \sum_{t'} ||\mathcal{M}_{\theta_{f,test}}(s_{t':t'+m}) - s_{t'+m:t'+m+n}||^2 \quad (10)$$

We prove Thm.2 by utilizing the triangular inequality and the prediction optimality of $\theta_{s,test}$ on synthetic data $s$. The complete proof is in App.A.2. Accoding to Thm.2, we obtain an optimizable upper bound of the value term. Therefore the optimization objective for the value term can be naturally defined as minimizing the upper bound of the value term, as shown in Eq.11.

$$\min_s \mathcal{L}_{label} \text{ where } \mathcal{L}_{label} = \sum_{t'} ||\mathcal{M}_{\theta_{f,test}}(s_{t':t'+m}) - s_{t'+m:t'+m+n}||^2 \quad (11)$$

According to Thm.2, label error $\mathcal{L}_{label}$ is the upper bound of the value term. Therefore, by minimizing the upper bound of the value term, the value term is indirectly minimized.

### 4.4 CondTSF

In this section, we develop a one-line plugin called CondTSF to minimize the label error $\mathcal{L}_{label}$ in Eq.11 so that the **value term** can be optimized. CondTSF is a lightweight one-line plugin, no backpropagation or gradient is required during the update. CondTSF utilizes a simple yet effective additive method to iteratively update the synthetic data $s$ and minimize the label error $\mathcal{L}_{label}$. In TS-forecasting, when generating training data, the data is usually sampled overlap from the dataset. Inspired by the overlap property, we utilize an additive method in CondTSF to gradually update the synthetic data to avoid vibrations. In the $i_{\text{th}}$ update iteration, CondTSF uses the prediction of expert model $\mathcal{M}_{f,test}(s_{t':t'+m})$ to update synthetic label $s_{t'+m:t'+m+n}$. The update process is shown in Eq.12.

$$s_{t'+m:t'+m+n}^{(i+1)} = (1-\beta) \cdot s_{t'+m:t'+m+n}^{(i)} + \beta \cdot \mathcal{M}_{\theta_{f,test}}(s_{t':t'+m}^{(i)}) \quad (12)$$

Table 1: Distill performance of different dataset condensation methods. For each method, ✗means CondTSF is not used, ✓means CondTSF is used, and ↓ means the decreased percentage of test error after CondTSF is applied. Five synthetic datasets are distilled and the average and standard deviation are reported.

| | CondTSF | ExchangeRate | | Weather | | Electricity | | Traffic | |
|---|---|---|---|---|---|---|---|---|---|
| | | MAE | MSE | MAE | MSE | MAE | MSE | MAE | MSE |
| Random | - | 0.783±0.090 | 1.070±0.246 | 0.530±0.084 | 0.647±0.159 | 0.840±0.017 | 1.102±0.031 | 0.854±0.018 | 1.350±0.043 |
| DC | ✗ | 0.716±0.090 | 0.875±0.217 | 0.483±0.053 | 0.530±0.087 | 0.808±0.017 | 1.017±0.046 | 0.823±0.007 | 1.296±0.021 |
| | ✓ | 0.602±0.115 | 0.632±0.215 | 0.449±0.055 | 0.467±0.084 | 0.794±0.014 | 0.987±0.035 | 0.818±0.012 | 1.265±0.032 |
| | ↓ | 15.8% | 27.8% | 6.9% | 11.7% | 1.7% | 2.9% | 0.7% | 2.4% |
| MTT | ✗ | 0.778±0.084 | 0.964±0.136 | 0.509±0.065 | 0.538±0.085 | 0.747±0.012 | 0.840±0.019 | 0.742±0.010 | 1.052±0.024 |
| | ✓ | 0.195±0.007 | 0.061±0.004 | 0.326±0.009 | 0.284±0.007 | 0.391±0.003 | 0.284±0.004 | 0.494±0.022 | 0.579±0.037 |
| | ↓ | 75.0% | 93.7% | 36.0% | 47.2% | 47.6% | 66.1% | 33.4% | 45.0% |
| PP | ✗ | 0.683±0.128 | 0.806±0.248 | 0.474±0.049 | 0.492±0.067 | 0.733±0.011 | 0.820±0.018 | 0.741±0.013 | 1.037±0.035 |
| | ✓ | 0.191±0.006 | 0.058±0.003 | 0.324±0.006 | 0.283±0.005 | 0.390±0.006 | 0.285±0.006 | 0.490±0.013 | 0.570±0.020 |
| | ↓ | 72.0% | 92.8% | 31.7% | 42.5% | 46.8% | 65.3% | 33.8% | 45.1% |
| TESLA | ✗ | 0.730±0.124 | 0.801±0.211 | 0.522±0.011 | 0.557±0.020 | 0.719±0.029 | 0.790±0.052 | 0.741±0.020 | 1.063±0.051 |
| | ✓ | 0.188±0.014 | 0.059±0.008 | 0.295±0.010 | 0.276±0.013 | 0.389±0.005 | 0.293±0.006 | 0.576±0.016 | 0.730±0.025 |
| | ↓ | 74.3% | 92.7% | 43.6% | 50.3% | 46.0% | 62.9% | 22.2% | 31.3% |
| FTD | ✗ | 0.690±0.153 | 0.818±0.278 | 0.511±0.037 | 0.535±0.048 | 0.748±0.012 | 0.844±0.019 | 0.745±0.007 | 1.054±0.014 |
| | ✓ | 0.184±0.005 | 0.055±0.003 | 0.320±0.005 | 0.280±0.004 | 0.396±0.003 | 0.290±0.002 | 0.501±0.021 | 0.587±0.032 |
| | ↓ | 73.3% | 93.3% | 37.3% | 47.6% | 47.1% | 65.6% | 32.7% | 44.3% |
| DATM | ✗ | 0.646±0.137 | 0.702±0.243 | 0.515±0.035 | 0.554±0.038 | 0.752±0.016 | 0.850±0.027 | 0.740±0.013 | 1.043±0.026 |
| | ✓ | 0.190±0.010 | 0.058±0.006 | 0.320±0.015 | 0.290±0.014 | 0.381±0.005 | 0.276±0.005 | 0.496±0.016 | 0.582±0.025 |
| | ↓ | 70.6% | 91.8% | 37.9% | 47.6% | 49.4% | 67.6% | 33.0% | 44.2% |
| Full | - | 0.110±0.001 | 0.023±0.000 | 0.197±0.001 | 0.131±0.001 | 0.312±0.002 | 0.223±0.002 | 0.406±0.003 | 0.492±0.004 |

| | CondTSF | ETTm1 | | ETTm2 | | ETTh1 | | ETTh2 | |
|---|---|---|---|---|---|---|---|---|---|
| | | MAE | MSE | MAE | MSE | MAE | MSE | MAE | MSE |
| Random | - | 0.728±0.033 | 0.993±0.082 | 0.695±0.011 | 0.889±0.030 | 0.756±0.035 | 1.059±0.083 | 0.749±0.037 | 1.013±0.089 |
| DC | ✗ | 0.672±0.020 | 0.859±0.038 | 0.631±0.023 | 0.708±0.063 | 0.704±0.053 | 0.933±0.118 | 0.627±0.081 | 0.694±0.158 |
| | ✓ | 0.661±0.012 | 0.833±0.018 | 0.591±0.026 | 0.603±0.044 | 0.678±0.034 | 0.873±0.070 | 0.601±0.027 | 0.631±0.060 |
| | ↓ | 1.8% | 3.1% | 6.4% | 14.9% | 3.6% | 6.4% | 4.1% | 9.1% |
| MTT | ✗ | 0.653±0.019 | 0.771±0.040 | 0.685±0.022 | 0.754±0.051 | 0.693±0.009 | 0.845±0.023 | 0.719±0.006 | 0.827±0.016 |
| | ✓ | 0.491±0.004 | 0.502±0.008 | 0.347±0.028 | 0.202±0.028 | 0.532±0.014 | 0.580±0.029 | 0.329±0.003 | 0.205±0.002 |
| | ↓ | 24.8% | 34.9% | 49.3% | 73.3% | 23.3% | 31.3% | 54.2% | 75.2% |
| PP | ✗ | 0.660±0.014 | 0.788±0.032 | 0.615±0.093 | 0.620±0.168 | 0.694±0.008 | 0.851±0.018 | 0.673±0.052 | 0.757±0.086 |
| | ✓ | 0.489±0.005 | 0.491±0.013 | 0.336±0.024 | 0.190±0.023 | 0.527±0.011 | 0.574±0.029 | 0.336±0.004 | 0.211±0.005 |
| | ↓ | 26.0% | 37.7% | 45.4% | 69.4% | 24.1% | 32.6% | 50.1% | 72.1% |
| TESLA | ✗ | 0.641±0.009 | 0.751±0.018 | 0.577±0.142 | 0.570±0.210 | 0.674±0.013 | 0.813±0.030 | 0.616±0.095 | 0.630±0.154 |
| | ✓ | 0.542±0.037 | 0.622±0.058 | 0.292±0.001 | 0.155±0.001 | 0.533±0.020 | 0.588±0.048 | 0.332±0.007 | 0.208±0.006 |
| | ↓ | 15.4% | 17.2% | 49.4% | 72.8% | 21.0% | 27.7% | 46.2% | 67.0% |
| FTD | ✗ | 0.663±0.009 | 0.790±0.020 | 0.563±0.147 | 0.571±0.221 | 0.693±0.016 | 0.857±0.044 | 0.625±0.148 | 0.686±0.240 |
| | ✓ | 0.494±0.007 | 0.502±0.010 | 0.347±0.012 | 0.200±0.012 | 0.529±0.014 | 0.570±0.030 | 0.335±0.009 | 0.210±0.008 |
| | ↓ | 25.5% | 36.5% | 38.4% | 65.0% | 23.7% | 33.4% | 46.5% | 69.4% |
| DATM | ✗ | 0.642±0.031 | 0.768±0.050 | 0.644±0.047 | 0.679±0.090 | 0.689±0.036 | 0.870±0.057 | 0.611±0.150 | 0.650±0.245 |
| | ✓ | 0.531±0.032 | 0.569±0.045 | 0.305±0.006 | 0.167±0.005 | 0.532±0.028 | 0.582±0.068 | 0.330±0.004 | 0.209±0.003 |
| | ↓ | 17.2% | 25.9% | 52.6% | 75.4% | 22.7% | 33.1% | 45.9% | 67.8% |
| Full | - | 0.432±0.001 | 0.473±0.001 | 0.230±0.001 | 0.113±0.001 | 0.389±0.003 | 0.339±0.004 | 0.276±0.002 | 0.166±0.002 |

where $0 < \beta < 1$ is the update ratio of this additive update method. This additive update process lowers the label error $\mathcal{L}_{label}$ of $s$ in each iteration exponentially, which can be formulated as

$$
\begin{aligned}
\mathcal{L}_{label}^{(i+1)} &= \sum_{t'} ||s_{t'+m:t'+m+n}^{(i+1)} - \mathcal{M}_{\theta_{f,test}}(s_{t':t'+m}^{(i+1)})||^2 \\
&= (1-\beta)^2 \sum_{t'} ||s_{t'+m:t'+m+n}^{(i)} - \mathcal{M}_{\theta_{f,test}}(s_{t':t'+m}^{(i)})||^2 = (1-\beta)^2 \mathcal{L}_{label}^{(i)}
\end{aligned}
\tag{13}
$$

Since the update ratio has a value of $0 < \beta < 1$, CondTSF lowers the label error $\mathcal{L}_{label}$ exponentially in each update iteration and solves the optimization problem for the value term. As a plugin module, CondTSF is used to update once for every $G$ iteration of parameter matching methods. In this way, the **gradient term** and the **value term** can be optimized synchronously. The algorithm is shown in Alg.1. We also formulate the complete condensation process using CondTSF, as shown in Fig.2.

## 5 Experiment

### 5.1 Experiment Settings

**Dataset Settings:** The efficacy of dataset condensation methods is assessed across eight time series datasets. For all datasets, the model is set to be using 24 steps of data to forecast 24 steps of data. We

Table 2: Information and condensation settings of time series datasets.

|  | ETTm1&ETTm2 | ETTh1&ETTh2 | ExchangeRate | Weather | Electricity | Traffic |
|---|---|---|---|---|---|---|
| Dataset length | 57600 | 14400 | 7588 | 52696 | 26304 | 17544 |
| Distill ratio | 0.83‰ | 3.33‰ | 6.33‰ | 0.91‰ | 1.82‰ | 2.74‰ |
| Distilled length | 48 | 48 | 48 | 48 | 48 | 48 |

Table 3: Generalization ability of different dataset condensation methods. For each dataset and each method, MLP, LSTM, CNN are trained with the synthetic data distilled from DLinear expert models. For each architecture, five test models are trained, the average and standard deviation of MAE, MSE are summarized. The result of CondTSF is using MTT as the backbone.

**ExchangeRate**

|  | MLP MAE | MLP MSE | LSTM MAE | LSTM MSE | CNN MAE | CNN MSE |
|---|---|---|---|---|---|---|
| Random | 0.931±0.024 | 1.246±0.057 | 0.840±0.047 | 1.035±0.102 | 0.910±0.038 | 1.217±0.106 |
| DC | 0.713±0.059 | 0.740±0.114 | 0.511±0.034 | 0.390±0.048 | 0.588±0.049 | 0.519±0.072 |
| KIP | 0.483±0.012 | 0.397±0.013 | 0.512±0.024 | 0.422±0.026 | 0.494±0.022 | 0.414±0.027 |
| FRePo | 0.564±0.033 | 0.537±0.041 | 0.583±0.048 | 0.569±0.077 | 0.599±0.025 | 0.578±0.044 |
| MTT | 0.421±0.007 | 0.301±0.009 | 0.431±0.010 | 0.313±0.009 | 0.419±0.007 | 0.300±0.010 |
| PP | 0.383±0.009 | 0.249±0.008 | 0.388±0.013 | 0.252±0.011 | 0.465±0.021 | 0.343±0.024 |
| TESLA | 0.316±0.008 | 0.172±0.007 | 0.323±0.010 | 0.175±0.007 | 0.439±0.034 | 0.302±0.044 |
| FTD | 0.425±0.007 | 0.306±0.005 | 0.433±0.012 | 0.310±0.011 | 0.445±0.025 | 0.329±0.031 |
| DATM | 0.452±0.013 | 0.349±0.016 | 0.229±0.045 | 0.095±0.032 | 0.351±0.053 | 0.209±0.049 |
| CondTSF | **0.135±0.005** | **0.032±0.002** | **0.135±0.004** | **0.032±0.002** | **0.248±0.031** | **0.101±0.022** |

**Weather**

|  | MLP MAE | MLP MSE | LSTM MAE | LSTM MSE | CNN MAE | CNN MSE |
|---|---|---|---|---|---|---|
| Random | 0.554±0.010 | 0.632±0.016 | 0.531±0.020 | 0.598±0.033 | 0.570±0.006 | 0.655±0.017 |
| DC | 0.503±0.014 | 0.540±0.022 | 0.446±0.011 | 0.430±0.017 | 0.517±0.016 | 0.533±0.028 |
| KIP | 0.293±0.008 | 0.276±0.011 | 0.262±0.004 | 0.253±0.004 | 0.331±0.005 | 0.292±0.003 |
| FRePo | 0.393±0.013 | 0.401±0.011 | 0.419±0.044 | 0.424±0.043 | 0.434±0.011 | 0.428±0.016 |
| MTT | 0.286±0.006 | 0.256±0.004 | 0.279±0.007 | 0.249±0.004 | 0.328±0.018 | 0.276±0.014 |
| PP | 0.279±0.019 | 0.253±0.010 | 0.309±0.009 | 0.271±0.008 | 0.344±0.023 | 0.315±0.031 |
| TESLA | 0.298±0.012 | 0.266±0.007 | 0.292±0.012 | 0.253±0.010 | 0.331±0.004 | 0.283±0.006 |
| FTD | 0.286±0.010 | 0.251±0.005 | 0.303±0.028 | 0.264±0.017 | 0.347±0.015 | 0.305±0.014 |
| DATM | 0.270±0.004 | 0.258±0.004 | 0.275±0.012 | 0.253±0.010 | 0.323±0.022 | 0.282±0.021 |
| CondTSF | **0.242±0.009** | **0.229±0.006** | **0.248±0.004** | **0.231±0.004** | **0.283±0.007** | **0.256±0.004** |

**Electricity**

|  | MLP MAE | MLP MSE | LSTM MAE | LSTM MSE | CNN MAE | CNN MSE |
|---|---|---|---|---|---|---|
| Random | 0.790±0.016 | 0.931±0.039 | 0.758±0.007 | 0.866±0.016 | 0.782±0.012 | 0.919±0.034 |
| DC | 0.778±0.007 | 0.912±0.016 | 0.770±0.004 | 0.884±0.009 | 0.769±0.010 | 0.897±0.022 |
| KIP | 0.769±0.014 | 0.881±0.029 | 0.700±0.018 | 0.741±0.036 | 0.761±0.016 | 0.864±0.035 |
| FRePo | 0.620±0.009 | 0.626±0.016 | 0.633±0.016 | 0.625±0.029 | 0.642±0.011 | 0.665±0.022 |
| MTT | 0.465±0.009 | 0.374±0.010 | 0.467±0.013 | 0.378±0.015 | 0.491±0.008 | 0.405±0.010 |
| PP | 0.483±0.008 | 0.388±0.009 | 0.481±0.008 | 0.388±0.009 | 0.521±0.015 | 0.444±0.021 |
| TESLA | 0.515±0.006 | 0.441±0.007 | 0.515±0.012 | 0.439±0.011 | 0.530±0.006 | 0.462±0.009 |
| FTD | 0.505±0.009 | 0.418±0.010 | 0.500±0.015 | 0.414±0.016 | 0.539±0.006 | 0.470±0.008 |
| DATM | 0.501±0.011 | 0.416±0.012 | 0.509±0.018 | 0.428±0.023 | 0.511±0.005 | 0.431±0.007 |
| CondTSF | **0.326±0.002** | **0.231±0.002** | **0.324±0.012** | **0.230±0.007** | **0.373±0.008** | **0.272±0.008** |

**Traffic**

|  | MLP MAE | MLP MSE | LSTM MAE | LSTM MSE | CNN MAE | CNN MSE |
|---|---|---|---|---|---|---|
| Random | 0.743±0.015 | 1.102±0.042 | 0.742±0.007 | 1.088±0.015 | 0.753±0.016 | 1.100±0.031 |
| DC | 0.730±0.011 | 1.035±0.031 | 0.709±0.012 | 0.989±0.030 | 0.747±0.009 | 1.068±0.020 |
| KIP | 0.738±0.018 | 1.056±0.045 | 0.714±0.017 | 1.008±0.023 | 0.753±0.018 | 1.074±0.023 |
| FRePo | 0.645±0.007 | 0.802±0.014 | 0.650±0.005 | 0.811±0.012 | 0.656±0.003 | 0.817±0.006 |
| MTT | 0.635±0.004 | 0.797±0.009 | 0.634±0.008 | 0.788±0.011 | 0.655±0.007 | 0.817±0.007 |
| PP | 0.617±0.006 | 0.751±0.006 | 0.610±0.008 | 0.740±0.010 | 0.593±0.004 | 0.745±0.010 |
| TESLA | 0.623±0.009 | 0.800±0.014 | 0.603±0.004 | 0.778±0.011 | 0.631±0.003 | 0.809±0.009 |
| FTD | 0.635±0.013 | 0.787±0.018 | 0.644±0.016 | 0.796±0.024 | 0.632±0.005 | 0.783±0.011 |
| DATM | 0.583±0.008 | 0.707±0.015 | 0.592±0.004 | 0.709±0.009 | 0.598±0.005 | 0.726±0.012 |
| CondTSF | **0.423±0.004** | **0.498±0.003** | **0.419±0.006** | **0.488±0.007** | **0.454±0.005** | **0.522±0.006** |

**ETTm1**

|  | MLP MAE | MLP MSE | LSTM MAE | LSTM MSE | CNN MAE | CNN MSE |
|---|---|---|---|---|---|---|
| Random | 0.697±0.009 | 0.859±0.020 | 0.677±0.017 | 0.801±0.033 | 0.713±0.015 | 0.891±0.027 |
| DC | 0.662±0.006 | 0.786±0.003 | 0.636±0.007 | 0.741±0.011 | 0.676±0.013 | 0.808±0.028 |
| KIP | 0.566±0.005 | 0.697±0.019 | 0.555±0.008 | 0.690±0.018 | 0.571±0.007 | 0.694±0.015 |
| FRePo | 0.599±0.007 | 0.718±0.013 | 0.611±0.028 | 0.738±0.048 | 0.630±0.031 | 0.749±0.086 |
| MTT | 0.484±0.003 | 0.484±0.005 | 0.515±0.033 | 0.530±0.048 | 0.563±0.006 | 0.608±0.019 |
| PP | 0.486±0.007 | 0.474±0.008 | 0.527±0.031 | 0.539±0.040 | 0.581±0.019 | 0.644±0.036 |
| TESLA | 0.519±0.003 | 0.523±0.003 | 0.513±0.007 | 0.516±0.007 | 0.579±0.013 | 0.620±0.020 |
| FTD | 0.528±0.015 | 0.579±0.032 | 0.631±0.015 | 0.790±0.037 | 0.576±0.024 | 0.626±0.041 |
| DATM | 0.499±0.007 | 0.516±0.006 | 0.513±0.015 | 0.524±0.016 | 0.577±0.030 | 0.616±0.046 |
| CondTSF | **0.452±0.004** | **0.455±0.001** | **0.459±0.013** | **0.461±0.011** | **0.520±0.018** | **0.543±0.025** |

**ETTm2**

|  | MLP MAE | MLP MSE | LSTM MAE | LSTM MSE | CNN MAE | CNN MSE |
|---|---|---|---|---|---|---|
| Random | 0.732±0.017 | 0.880±0.041 | 0.754±0.020 | 0.927±0.054 | 0.760±0.021 | 0.934±0.056 |
| DC | 0.623±0.006 | 0.629±0.069 | 0.532±0.028 | 0.459±0.048 | 0.682±0.023 | 0.745±0.051 |
| KIP | 0.285±0.012 | 0.144±0.009 | 0.290±0.021 | 0.149±0.015 | 0.347±0.031 | 0.201±0.028 |
| FRePo | 0.476±0.032 | 0.412±0.043 | 0.472±0.057 | 0.395±0.089 | 0.579±0.075 | 0.574±0.142 |
| MTT | 0.258±0.007 | 0.129±0.005 | 0.246±0.005 | 0.124±0.004 | 0.340±0.016 | 0.193±0.017 |
| PP | 0.272±0.004 | 0.136±0.003 | 0.269±0.002 | 0.135±0.002 | 0.308±0.013 | 0.167±0.010 |
| TESLA | 0.272±0.004 | 0.135±0.003 | 0.272±0.007 | 0.135±0.006 | 0.365±0.041 | 0.221±0.043 |
| FTD | 0.279±0.005 | 0.142±0.004 | 0.290±0.011 | 0.147±0.010 | 0.403±0.025 | 0.254±0.024 |
| DATM | 0.293±0.004 | 0.153±0.004 | 0.290±0.009 | 0.149±0.007 | 0.377±0.030 | 0.232±0.029 |
| CondTSF | **0.231±0.002** | **0.107±0.001** | **0.240±0.009** | **0.111±0.005** | **0.273±0.021** | **0.133±0.014** |

**ETTh1**

|  | MLP MAE | MLP MSE | LSTM MAE | LSTM MSE | CNN MAE | CNN MSE |
|---|---|---|---|---|---|---|
| Random | 0.670±0.011 | 0.796±0.023 | 0.658±0.012 | 0.773±0.022 | 0.703±0.014 | 0.874±0.039 |
| DC | 0.643±0.019 | 0.745±0.038 | 0.626±0.014 | 0.718±0.013 | 0.672±0.023 | 0.802±0.041 |
| KIP | 0.636±0.017 | 0.732±0.029 | 0.608±0.016 | 0.696±0.027 | 0.650±0.016 | 0.758±0.021 |
| FRePo | 0.653±0.004 | 0.770±0.007 | 0.640±0.009 | 0.754±0.019 | 0.659±0.010 | 0.783±0.025 |
| MTT | 0.606±0.003 | 0.673±0.009 | 0.613±0.005 | 0.680±0.010 | 0.612±0.007 | 0.692±0.011 |
| PP | 0.633±0.006 | 0.719±0.006 | 0.630±0.006 | 0.710±0.009 | 0.635±0.007 | 0.730±0.007 |
| TESLA | 0.602±0.005 | 0.671±0.010 | 0.590±0.005 | 0.651±0.013 | 0.612±0.005 | 0.691±0.015 |
| FTD | 0.616±0.008 | 0.710±0.014 | 0.618±0.008 | 0.716±0.011 | 0.626±0.008 | 0.725±0.014 |
| DATM | 0.617±0.004 | 0.681±0.010 | 0.612±0.007 | 0.672±0.015 | 0.637±0.004 | 0.723±0.010 |
| CondTSF | **0.434±0.001** | **0.397±0.001** | **0.429±0.002** | **0.388±0.005** | **0.473±0.006** | **0.456±0.008** |

**ETTh2**

|  | MLP MAE | MLP MSE | LSTM MAE | LSTM MSE | CNN MAE | CNN MSE |
|---|---|---|---|---|---|---|
| Random | 0.732±0.012 | 0.874±0.031 | 0.702±0.014 | 0.799±0.031 | 0.755±0.027 | 0.908±0.062 |
| DC | 0.619±0.019 | 0.650±0.043 | 0.534±0.019 | 0.481±0.036 | 0.680±0.028 | 0.746±0.053 |
| KIP | 0.494±0.009 | 0.419±0.011 | 0.431±0.012 | 0.329±0.016 | 0.551±0.032 | 0.492±0.051 |
| FRePo | 0.570±0.033 | 0.552±0.060 | 0.485±0.036 | 0.405±0.054 | 0.672±0.023 | 0.728±0.054 |
| MTT | 0.307±0.005 | 0.182±0.004 | 0.305±0.014 | 0.180±0.010 | 0.374±0.033 | 0.246±0.036 |
| PP | 0.354±0.019 | 0.229±0.020 | 0.292±0.012 | 0.173±0.007 | 0.450±0.060 | 0.346±0.085 |
| TESLA | 0.308±0.007 | 0.181±0.006 | 0.292±0.004 | 0.170±0.003 | 0.390±0.016 | 0.257±0.014 |
| FTD | 0.329±0.003 | 0.197±0.004 | 0.312±0.007 | 0.187±0.012 | 0.386±0.014 | 0.249±0.023 |
| DATM | 0.337±0.005 | 0.208±0.006 | 0.329±0.006 | 0.200±0.006 | 0.398±0.015 | 0.268±0.013 |
| CondTSF | **0.290±0.005** | **0.168±0.004** | **0.287±0.006** | **0.166±0.004** | **0.342±0.008** | **0.211±0.006** |

set the length of the synthetic dataset as 48, as shown in Table.2. Each synthetic dataset can only generate one training pair. We conduct experiments with two larger distill ratios as shown in App.B.

**Model Settings:** We plug CondTSF into existing dataset condensation models based on parameter matching, including DC[51], MTT[3], PP[21], TESLA[5], FTD[7] and DATM[11] to prove the effectiveness of CondTSF. We also conduct experiments on non-parameter-matching based methods, including DM[50], IDM[52], KIP[33], FRePo[55] to prove that optimizing value term only also helps boost the performance. The experiment setting and results are shown in App.E. We use DLinear[45] as the expert model to perform dataset condensation since DLinear is a linear model.

**Metric Settings:** The source dataset is first divided into a train set and a test set. All synthetic data is initialized by randomly sampling data from the train set. After a synthetic dataset is finished distilling, it is used to train another five models. After the five models are trained, they are tested on the test set. Their average mean absolute error (MAE) and mean square error (MSE) are recorded. We repeat the process above five times and report the average and standard deviation. While testing the generalization ability of the dataset condensation methods, DLinear[45] is used as the expert model to perform dataset condensation. Meanwhile, MLP, LSTM[13], and CNN are used as test models when testing the generalization ability of the dataset condensation methods.

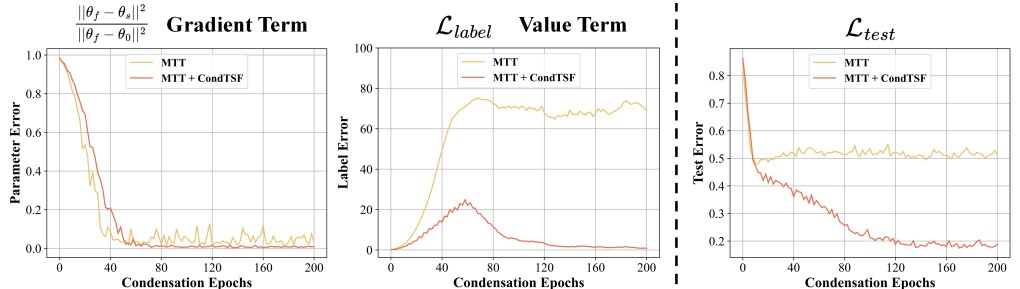

Figure 4: Changing trajectory of **Left:** parameter error which refer to gradient term, **Middle:** label error which refer to value term and **Right:** test error during dataset condensation process.

**Implementation Details:** As a plugin module, we test CondTSF with all previous methods. Each synthetic dataset is optimized using a standard training process according to the chosen backbone model. CondTSF is set to update every 3 epochs and the additive update ratio $\beta$ is set to be 0.01. All the experiments are carried out on an NVIDIA RTX 3080Ti.

## 5.2 Results

**Single Architecture Performance:** The results are summarized in Table.1. For each backbone method, the first line shows the performance of the backbone model, the second line shows the performance of a backbone model with CondTSF, and the third line shows the percentage of reduction in MAE and MSE after CondTSF is applied. There's a considerable reduction in error for all backbone models. The results suggest that CondTSF is effective in optimizing the value term and enhancing the performance in dataset condensation for TS-forecasting. However, using CondTSF on DC[51] is not as effective as other methods. The reason is that instead of directly matching parameters, DC matches the gradient of parameters on loss in each iteration. Indirectly matching gradient leads to accumulating errors in parameters, making DC unable to lower parameter error as effectively as directly matching parameters. Therefore CondTSF is not effective enough when applied to DC[51].

**Cross Architecture Performance:** We also conduct experiments to evaluate the cross-architecture performance of dataset condensation methods. The results are summarized in Table.3. We test all models on all datasets with MLP, LSTM[13], and CNN as test models. All synthetic data is distilled using DLinear[45] model as experts. We use MTT[3] as the backbone for CondTSF. We observe that CondTSF based on MTT outperformed all other previous models.

## 5.3 Discussion

**Test Performance and Errors:** We conduct experiments on ExchangeRate dataset with MTT[3] and MTT+CondTSF. As shown in Fig.4, trajectory of parameter error $\frac{||\theta_f - \theta_s||^2}{||\theta_f - \theta_0||^2}$, label error $\mathcal{L}_{label}$ and test error $\mathcal{L}_{test}$ through the distillation process are presented. Regarding the parameter error corresponding to the gradient term, both MTT and MTT+CondTSF converge quickly, suggesting that the incorporation of CondTSF doesn't impact parameter alignment. As for the label error corresponding to the value term, since the initial synthetic data $s$ is randomly sampled from the train set $f$ and the expert model is trained by the train set $f$, the label error of $s$ is small at the beginning. However, the utilization of MTT results in an elevation of label error, whereas employing CondTSF effectively mitigates this increase in label error. During the test, MTT+CondTSF notably outperforms MTT by concurrently optimizing both the value term and the gradient term.

## 6 Limitations

The limitation of this work is that we use linear models in our analysis so that the gradient of a model on input is the parameter of the model. Therefore, only linear models like DLinear[45] are solid enough to be an expert model for dataset condensation. The analysis no longer holds when it comes to more complicated models. However, experiments in App.D and App.E show that CondTSF is

also effective with non-parameter-matching methods and non-linear models, which merits further exploration.

## 7 Conclusion

In this study, we provide abundant proof that previous dataset condensation methods based on classification are not suitable for dataset condensation for TS-forecasting. We elucidate that these earlier methods, predominantly focused on classification tasks, only address a portion of the optimization objective pertinent to TS-forecasting. To address this issue, we propose a plugin module called CondTSF that can collaborate with parameter matching based dataset condensation methods. CondTSF optimizes the optimization objective that previous methods have neglected and boosts the performance of dataset condensation methods on TS-forecasting. We conduct experiments on eight widely used time series datasets and prove the effectiveness of our proof and method. CondTSF consistently enhances the performance of all previous techniques across all datasets, substantiating its effectiveness in improving dataset condensation outcomes for TS-forecasting applications.

## Acknowledgements

This work was sponsored by National Natural Science Foundation of China under Grant No. 62102246, 62272301, and Provincial Key Research and Development Program of Zhejiang under Grant No. 2021C01034. Part of the work was done when the students were doing internships at Yunqi Academy of Engineering.

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

# A Complete Proof

## A.1 Complete Proof for Theorem 1

**Theorem 1.** *Given arbitrary synthetic data $s_{t':t'+m}$, the upper bound of the optimization objective of dataset condensation $\mathcal{L}_{test}(\mathcal{M}_{\theta_{s,test}}, x)$ can be formulated as such*

$$\mathcal{L}_{test}(\mathcal{M}_{\theta_{s,test}}, x) \leq \sum_t ||\epsilon||^2 + \underbrace{||\mathcal{M}_{\theta_{s,test}}(s_{t':t'+m}) - \mathcal{M}_{\theta_{f,test}}(s_{t':t'+m})||^2}_{\textbf{\textit{Value Term}}}$$
$$+ \underbrace{||(\nabla \mathcal{M}_{\theta_{s,test}}(s_{t':t'+m}) - \nabla \mathcal{M}_{\theta_{f,test}}(s_{t':t'+m}))^\top (x_{t:t+m} - s_{t':t'+m})||^2}_{\textbf{\textit{Gradient Term}}} \quad (14)$$

*Proof.* Replacing the true label $x_{t+m:t+m+n}$ in $\mathcal{L}_{test}(\mathcal{M}_{\theta_{s,test}}, x)$ with Eq.4, the optimization objective of dataset condensation for TS-forecasting is reformulated as the distance between the predictions of $\mathcal{M}_{\theta_{s,test}}$ and $\mathcal{M}_{\theta_{f,test}}$ given the same test input. Then the triangular inequality of norm functions is used and the original optimization objective can be transformed to its upper bound, as shown in Eq.15.

$$\mathcal{L}_{test}(\mathcal{M}_{\theta_{s,test}}, x) = \sum_t ||\mathcal{M}_{\theta_{s,test}}(x_{t:t+m}) - x_{t+m:t+m+n}||^2$$
$$= \sum_t ||\mathcal{M}_{\theta_{s,test}}(x_{t:t+m}) - \mathcal{M}_{\theta_{f,test}}(x_{t:t+m}) - \epsilon||^2 \quad (15)$$
$$\leq \sum_t ||\mathcal{M}_{\theta_{s,test}}(x_{t:t+m}) - \mathcal{M}_{\theta_{f,test}}(x_{t:t+m})||^2 + ||\epsilon||^2$$

In Eq.15, we prove that minimizing the distance between $\mathcal{M}_{\theta_{s,test}}(x_{t:t+m})$ and $\mathcal{M}_{\theta_{f,test}}(x_{t:t+m})$ is equivalent to minimizing the upper bound of the original optimization objective. Then we decompose the distance between predictions of $\mathcal{M}_{\theta_{s,test}}$ and $\mathcal{M}_{\theta_{f,test}}$ into two optimizable terms for better optimization. We use linear models for further analysis since linear models can be both effective and efficient in TS-forecasting tasks[45]. Given a linear model $\mathcal{M}_\theta(x) = \theta x$, its second and higher order gradient is zero, i.e. $\nabla^k \mathcal{M}_\theta(x) = 0, \forall k \geq 2$. Therefore, first-order Taylor Expansion can be utilized to get the prediction of the model $\mathcal{M}_\theta$ on test data $x_{t:t+m}$ using the prediction and gradient of the model $\mathcal{M}_\theta$ on arbitrary synthetic data $s_{t':t'+m}$. The process is formulated in Eq.16. Meanwhile, it is worth mentioning that if $\mathcal{M}_\theta$ is a non-linear model, then the second and higher-order terms of the Taylor Expansion are ignored in Eq.16.

$$\mathcal{M}_\theta(x_{t:t+m}) = \mathcal{M}_\theta(s_{t':t'+m}) + \nabla \mathcal{M}_\theta(s_{t':t'+m})^\top (x_{t:t+m} - s_{t':t'+m}) \quad (16)$$

Then we expand Eq.15 with Taylor expansion. After that, the triangular inequality of norm functions is used to get its upper bound. In the meantime, by applying the triangular inequality, the optimization objective can be decomposed into two optimizable terms.

$$\mathcal{L}_{test}(\mathcal{M}_{\theta_{s,test}}, x) \leq \sum_t ||\mathcal{M}_{\theta_{s,test}}(x_{t:t+m}) - \mathcal{M}_{\theta_{f,test}}(x_{t:t+m})||^2 + ||\epsilon||^2$$
$$= \sum_t ||\epsilon||^2 + ||\mathcal{M}_{\theta_{s,test}}(s_{t':t'+m}) + \nabla \mathcal{M}_{\theta_{s,test}}(s_{t':t'+m})^\top (x_{t:t+m} - s_{t':t'+m})$$
$$- \mathcal{M}_{\theta_{f,test}}(s_{t':t'+m}) - \nabla \mathcal{M}_{\theta_{f,test}}(s_{t':t'+m})^\top (x_{t:t+m} - s_{t':t'+m})||^2$$
$$= \sum_t ||\epsilon||^2 + ||\mathcal{M}_{\theta_{s,test}}(s_{t':t'+m}) - \mathcal{M}_{\theta_{f,test}}(s_{t':t'+m})$$
$$+ (\nabla \mathcal{M}_{\theta_{s,test}}(s_{t':t'+m}) - \nabla \mathcal{M}_{\theta_{f,test}}(s_{t':t'+m}))^\top (x_{t:t+m} - s_{t':t'+m})||^2$$
$$\leq \sum_t ||\epsilon||^2 + \underbrace{||\mathcal{M}_{\theta_{s,test}}(s_{t':t'+m}) - \mathcal{M}_{\theta_{f,test}}(s_{t':t'+m})||^2}_{\textbf{\textit{Value Term}}}$$
$$+ \underbrace{||(\nabla \mathcal{M}_{\theta_{s,test}}(s_{t':t'+m}) - \nabla \mathcal{M}_{\theta_{f,test}}(s_{t':t'+m}))^\top (x_{t:t+m} - s_{t':t'+m})||^2}_{\textbf{\textit{Gradient Term}}}$$

$$(17)$$

Therefore Thm.1 is proved. $\square$

## A.2 Complete Proof for Theorem 2

**Theorem 2.** *The upper bound of the value term can be formulated as such*

$$||\mathcal{M}_{\theta_{s,test}}(s_{t':t'+m}) - \mathcal{M}_{\theta_{f,test}}(s_{t':t'+m})||^2 \leq 2 \cdot \sum_{t'} ||\mathcal{M}_{\theta_{f,test}}(s_{t':t'+m}) - s_{t'+m:t'+m+n}||^2 \quad (18)$$

*Proof.* We first use triangular inequality and the non-negativity of norm functions to get the upper bound of the value term. The process is shown in Eq.19.

$$
\begin{aligned}
& ||\mathcal{M}_{\theta_{s,test}}(s_{t':t'+m}) - \mathcal{M}_{\theta_{f,test}}(s_{t':t'+m})||^2 \text{ (Value Term)} \\
= & ||\mathcal{M}_{\theta_{s,test}}(s_{t':t'+m}) - s_{t'+m:t'+m+n} + s_{t'+m:t'+m+n} - \mathcal{M}_{\theta_{f,test}}(s_{t':t'+m})||^2 \\
\leq & ||\mathcal{M}_{\theta_{s,test}}(s_{t':t'+m}) - s_{t'+m:t'+m+n}||^2 + ||s_{t'+m:t'+m+n} - \mathcal{M}_{\theta_{f,test}}(s_{t':t'+m})||^2 \\
\leq & \sum_{t'} ||\mathcal{M}_{\theta_{s,test}}(s_{t':t'+m}) - s_{t'+m:t'+m+n}||^2 + \sum_{t'} ||s_{t'+m:t'+m+n} - \mathcal{M}_{\theta_{f,test}}(s_{t':t'+m})||^2
\end{aligned}
$$
$$(19)$$

For further analysis, we need to step back and formulate the training process of $\theta_s$ to derive an inequality. By doing dataset condensation, a synthetic dataset $s$ is obtained. Then we formulate the training process of $\theta_{s,test}$ on synthetic data $s$ as minimizing the prediction error on $s$. The training process is formulated in Eq.20.

$$\theta_{s,test} = \arg\min_{\theta} \sum_{t'} ||\mathcal{M}_{\theta}(s_{t':t'+m}) - s_{t'+m:t'+m+n}||^2 \quad (20)$$

Now we can derive an inequality. We denote $s$ as the synthetic dataset obtained by dataset condensation. We denote $\mathcal{M}_{\theta_{s,test}}$ as the model that is trained on $s$ as shown Eq.20. According to Eq.20, $\mathcal{M}_{\theta_{s,test}}$ has the lowest prediction error on synthetic data $s$ under the given model architecture. Since $\mathcal{M}_{\theta_{s,test}}$ and $\mathcal{M}_{\theta_{f,test}}$ share the same model architecture, the prediction error of $\mathcal{M}_{\theta_{s,test}}$ on synthetic data $s$ is no larger than the prediction error of $\mathcal{M}_{f,test}$ on synthetic data $s$. This inequality can be formulated as such

$$\sum_{t'} ||\mathcal{M}_{\theta_{s,test}}(s_{t':t'+m}) - s_{t'+m:t'+m+n}||^2 \leq \sum_{t'} ||s_{t'+m:t'+m+n} - \mathcal{M}_{\theta_{f,test}}(s_{t':t'+m})||^2 \quad (21)$$

By applying Eq.21 to Eq.19, we obtain the upper bound of the value term, as shown in Eq.22.

$$
\begin{aligned}
& ||\mathcal{M}_{\theta_{s,test}}(s_{t':t'+m}) - \mathcal{M}_{\theta_{f,test}}(s_{t':t'+m})||^2 \text{ (Value Term)} \\
\leq & \sum_{t'} ||\mathcal{M}_{\theta_{s,test}}(s_{t':t'+m}) - s_{t'+m:t'+m+n}||^2 + \sum_{t'} ||s_{t'+m:t'+m+n} - \mathcal{M}_{\theta_{f,test}}(s_{t':t'+m})||^2 \\
\leq & 2 \cdot \sum_{t'} ||\mathcal{M}_{\theta_{f,test}}(s_{t':t'+m}) - s_{t'+m:t'+m+n}||^2
\end{aligned}
$$
$$(22)$$

Therefore Thm.2 is proved. □

# B  Performance with Different Distill Ratio

We further explore the performance of CondTSF with different distill ratios and compare the results with previous matching-based methods.

## B.1  Standard Ratio Condensation

We distill the dataset into a synthetic dataset with a flexible length for each dataset. The information on condensation in Table.4. The performance is shown in Table.5.

Table 4: Information and condensation settings of time series datasets.

|  | ETTm1&ETTm2 | ETTh1&ETTh2 | ExchangeRate | Weather | Electricity | Traffic |
|---|---|---|---|---|---|---|
| Dataset length | 57600 | 14400 | 7588 | 52696 | 26304 | 17544 |
| Distill ratio | 0.2% | 0.4% | 1% | 0.2% | 0.3% | 0.4% |
| Distilled length | 115 | 57 | 75 | 105 | 78 | 70 |

Table 5: Distill performance of different dataset condensation methods. For each method, ✗means CondTSF is not used, ✓means CondTSF is used, and ↓ means the decreased percentage of test error after CondTSF is applied. Five synthetic datasets are distilled and the average and standard deviation are reported.

| | CondTSF | ExchangeRate | | Weather | | Electricity | | Traffic | |
|---|---|---|---|---|---|---|---|---|---|
| | | MAE | MSE | MAE | MSE | MAE | MSE | MAE | MSE |
| Random | - | 0.730±0.168 | 0.957±0.397 | 0.566±0.056 | 0.708±0.135 | 0.832±0.024 | 1.080±0.058 | 0.845±0.007 | 1.343±0.017 |
| DC | ✗ | 0.657±0.025 | 0.729±0.062 | 0.488±0.006 | 0.523±0.007 | 0.797±0.014 | 0.973±0.031 | 0.816±0.007 | 1.257±0.023 |
| | ✓ | 0.645±0.014 | 0.710±0.039 | 0.450±0.041 | 0.469±0.069 | 0.769±0.041 | 0.920±0.104 | 0.810±0.017 | 1.250±0.039 |
| | ↓ | 1.9% | 2.6% | 7.7% | 10.4% | 3.6% | 5.5% | 0.7% | 0.5% |
| MTT | ✗ | 0.467±0.018 | 0.361±0.024 | 0.330±0.020 | 0.291±0.020 | 0.473±0.014 | 0.379±0.017 | 0.575±0.022 | 0.726±0.016 |
| | ✓ | 0.180±0.008 | 0.053±0.004 | 0.285±0.010 | 0.253±0.005 | 0.335±0.002 | 0.238±0.001 | 0.429±0.006 | 0.500±0.007 |
| | ↓ | 61.6% | 85.3% | 13.4% | 13.1% | 29.1% | 37.4% | 25.5% | 31.2% |
| PP | ✗ | 0.463±0.032 | 0.352±0.042 | 0.340±0.022 | 0.301±0.022 | 0.471±0.008 | 0.375±0.009 | 0.582±0.012 | 0.714±0.014 |
| | ✓ | 0.179±0.006 | 0.053±0.004 | 0.279±0.005 | 0.248±0.005 | 0.336±0.003 | 0.240±0.001 | 0.423±0.005 | 0.490±0.007 |
| | ↓ | 61.2% | 84.9% | 17.7% | 17.6% | 28.5% | 35.9% | 27.4% | 31.3% |
| TESLA | ✗ | 0.406±0.026 | 0.275±0.038 | 0.334±0.009 | 0.292±0.008 | 0.530±0.007 | 0.463±0.008 | 0.650±0.018 | 0.855±0.044 |
| | ✓ | 0.185±0.014 | 0.056±0.008 | 0.292±0.009 | 0.262±0.005 | 0.369±0.002 | 0.273±0.002 | 0.511±0.012 | 0.614±0.020 |
| | ↓ | 54.5% | 79.6% | 12.4% | 10.2% | 30.5% | 41.1% | 21.4% | 28.2% |
| FTD | ✗ | 0.445±0.038 | 0.332±0.050 | 0.324±0.010 | 0.284±0.014 | 0.470±0.003 | 0.374±0.004 | 0.557±0.016 | 0.680±0.008 |
| | ✓ | 0.173±0.003 | 0.049±0.002 | 0.274±0.006 | 0.246±0.004 | 0.329±0.004 | 0.232±0.003 | 0.410±0.005 | 0.476±0.004 |
| | ↓ | 61.2% | 85.1% | 15.3% | 13.4% | 30.0% | 38.1% | 26.4% | 30.0% |
| DATM | ✗ | 0.454±0.030 | 0.345±0.047 | 0.315±0.002 | 0.279±0.001 | 0.495±0.005 | 0.410±0.006 | 0.583±0.017 | 0.722±0.033 |
| | ✓ | 0.182±0.003 | 0.054±0.002 | 0.296±0.011 | 0.264±0.007 | 0.325±0.003 | 0.228±0.002 | 0.410±0.006 | 0.473±0.005 |
| | ↓ | 59.9% | 84.4% | 5.8% | 5.5% | 34.3% | 44.3% | 29.7% | 34.5% |
| Full | - | 0.110±0.001 | 0.023±0.000 | 0.197±0.001 | 0.131±0.001 | 0.312±0.002 | 0.223±0.002 | 0.406±0.003 | 0.492±0.004 |

| | CondTSF | ETTm1 | | ETTm2 | | ETTh1 | | ETTh2 | |
|---|---|---|---|---|---|---|---|---|---|
| | | MAE | MSE | MAE | MSE | MAE | MSE | MAE | MSE |
| Random | - | 0.697±0.054 | 0.934±0.105 | 0.629±0.129 | 0.747±0.285 | 0.725±0.067 | 0.995±0.152 | 0.645±0.118 | 0.763±0.251 |
| DC | ✗ | 0.665±0.012 | 0.837±0.024 | 0.575±0.015 | 0.574±0.030 | 0.713±0.024 | 0.933±0.049 | 0.591±0.069 | 0.619±0.132 |
| | ✓ | 0.659±0.010 | 0.828±0.021 | 0.542±0.078 | 0.516±0.138 | 0.695±0.019 | 0.901±0.039 | 0.488±0.092 | 0.429±0.148 |
| | ↓ | 0.8% | 1.1% | 5.7% | 10.0% | 2.5% | 3.5% | 17.5% | 30.8% |
| MTT | ✗ | 0.486±0.016 | 0.478±0.021 | 0.326±0.013 | 0.183±0.013 | 0.639±0.020 | 0.748±0.040 | 0.564±0.116 | 0.551±0.172 |
| | ✓ | 0.470±0.003 | 0.470±0.005 | 0.273±0.010 | 0.133±0.007 | 0.453±0.009 | 0.422±0.016 | 0.324±0.003 | 0.197±0.003 |
| | ↓ | 3.4% | 1.6% | 16.3% | 27.2% | 29.1% | 43.6% | 42.6% | 64.2% |
| PP | ✗ | 0.492±0.014 | 0.485±0.023 | 0.327±0.017 | 0.185±0.016 | 0.654±0.011 | 0.765±0.024 | 0.543±0.123 | 0.517±0.193 |
| | ✓ | 0.466±0.003 | 0.470±0.005 | 0.263±0.006 | 0.127±0.004 | 0.454±0.003 | 0.421±0.006 | 0.335±0.002 | 0.209±0.003 |
| | ↓ | 5.3% | 3.1% | 19.5% | 31.0% | 30.5% | 45.0% | 38.4% | 59.5% |
| TESLA | ✗ | 0.530±0.007 | 0.555±0.002 | 0.315±0.005 | 0.172±0.004 | 0.641±0.009 | 0.748±0.020 | 0.548±0.106 | 0.519±0.158 |
| | ✓ | 0.514±0.010 | 0.554±0.021 | 0.289±0.005 | 0.152±0.003 | 0.507±0.008 | 0.524±0.019 | 0.334±0.009 | 0.209±0.010 |
| | ↓ | 3.1% | 0.3% | 8.4% | 11.8% | 20.9% | 30.0% | 39.1% | 59.7% |
| FTD | ✗ | 0.490±0.006 | 0.476±0.010 | 0.330±0.017 | 0.186±0.017 | 0.633±0.011 | 0.730±0.021 | 0.611±0.038 | 0.622±0.064 |
| | ✓ | 0.463±0.005 | 0.466±0.003 | 0.264±0.007 | 0.128±0.005 | 0.427±0.003 | 0.379±0.006 | 0.313±0.004 | 0.186±0.003 |
| | ↓ | 5.4% | 2.1% | 19.9% | 31.3% | 32.6% | 48.0% | 48.8% | 70.1% |
| DATM | ✗ | 0.514±0.012 | 0.520±0.015 | 0.323±0.005 | 0.179±0.004 | 0.623±0.029 | 0.722±0.054 | 0.537±0.111 | 0.501±0.175 |
| | ✓ | 0.498±0.007 | 0.497±0.009 | 0.281±0.007 | 0.141±0.006 | 0.423±0.004 | 0.372±0.005 | 0.303±0.003 | 0.175±0.003 |
| | ↓ | 3.2% | 4.4% | 13.0% | 21.5% | 32.0% | 48.4% | 43.6% | 65.2% |
| Full | - | 0.432±0.001 | 0.473±0.001 | 0.230±0.001 | 0.113±0.001 | 0.389±0.003 | 0.339±0.004 | 0.276±0.002 | 0.166±0.002 |

## B.2   3-times Standard Ratio Condensation

We distill the dataset into a synthetic dataset with a flexible length for each dataset. Each synthetic dataset is 3 times larger than the synthetic data in Table.4. The information on condensation is shown in Table.6. The performance is shown in Table.7.

Table 6: Information and condensation settings of time series datasets.

|  | ETTm1&ETTm2 | ETTh1&ETTh2 | ExchangeRate | Weather | Electricity | Traffic |
|---|---|---|---|---|---|---|
| Dataset length | 57600 | 14400 | 7588 | 52696 | 26304 | 17544 |
| Distill ratio | 0.6% | 1.2% | 3% | 0.6% | 0.9% | 1.2% |
| Distilled length | 345 | 172 | 227 | 316 | 236 | 210 |

Table 7: Distill performance of different dataset condensation methods. For each method, ✗means CondTSF is not used, ✓means CondTSF is used, and ↓ means the decreased percentage of test error after CondTSF is applied. Five synthetic datasets are distilled and the average and standard deviation are reported.

|  | CondTSF | ExchangeRate | | Weather | | Electricity | | Traffic | |
|---|---|---|---|---|---|---|---|---|---|
|  |  | MAE | MSE | MAE | MSE | MAE | MSE | MAE | MSE |
| Random | - | 0.852±0.081 | 1.253±0.223 | 0.447±0.067 | 0.471±0.105 | 0.832±0.016 | 1.079±0.041 | 0.840±0.021 | 1.320±0.052 |
| DC | ✗ | 0.711±0.028 | 0.864±0.063 | 0.439±0.027 | 0.444±0.035 | 0.827±0.008 | 1.068±0.018 | 0.833±0.006 | 1.304±0.037 |
|  | ✓ | 0.614±0.117 | 0.658±0.224 | 0.396±0.013 | 0.372±0.019 | 0.804±0.003 | 1.012±0.009 | 0.816±0.005 | 1.271±0.003 |
|  | ↓ | 13.6% | 23.8% | 9.8% | 16.3% | 2.8% | 5.2% | 2.0% | 2.6% |
| MTT | ✗ | 0.201±0.012 | 0.066±0.008 | 0.324±0.023 | 0.293±0.027 | 0.332±0.004 | 0.242±0.003 | 0.432±0.007 | 0.520±0.008 |
|  | ✓ | 0.175±0.006 | 0.050±0.002 | 0.274±0.013 | 0.255±0.007 | 0.331±0.003 | 0.241±0.003 | 0.422±0.006 | 0.505±0.003 |
|  | ↓ | 12.8% | 23.7% | 15.6% | 12.7% | 0.3% | 0.1% | 2.3% | 2.9% |
| PP | ✗ | 0.198±0.008 | 0.064±0.005 | 0.308±0.015 | 0.277±0.015 | 0.333±0.004 | 0.242±0.003 | 0.435±0.005 | 0.522±0.008 |
|  | ✓ | 0.176±0.003 | 0.051±0.002 | 0.274±0.006 | 0.259±0.002 | 0.330±0.001 | 0.239±0.002 | 0.429±0.004 | 0.512±0.005 |
|  | ↓ | 11.0% | 19.9% | 11.0% | 6.5% | 1.0% | 1.2% | 1.4% | 1.8% |
| TESLA | ✗ | 0.209±0.016 | 0.071±0.011 | 0.297±0.005 | 0.265±0.003 | 0.446±0.011 | 0.371±0.014 | 0.593±0.011 | 0.734±0.023 |
|  | ✓ | 0.176±0.009 | 0.051±0.005 | 0.287±0.005 | 0.262±0.004 | 0.413±0.007 | 0.336±0.009 | 0.551±0.018 | 0.664±0.044 |
|  | ↓ | 15.6% | 27.9% | 3.6% | 1.1% | 7.3% | 9.3% | 7.1% | 9.5% |
| FTD | ✗ | 0.198±0.008 | 0.064±0.005 | 0.328±0.015 | 0.298±0.017 | 0.333±0.006 | 0.243±0.004 | 0.435±0.003 | 0.523±0.005 |
|  | ✓ | 0.172±0.004 | 0.049±0.002 | 0.281±0.007 | 0.258±0.004 | 0.331±0.005 | 0.243±0.004 | 0.421±0.003 | 0.501±0.005 |
|  | ↓ | 13.3% | 23.0% | 14.3% | 13.4% | 0.7% | 0.0% | 3.3% | 4.3% |
| DATM | ✗ | 0.196±0.010 | 0.062±0.005 | 0.284±0.009 | 0.264±0.008 | 0.335±0.006 | 0.244±0.005 | 0.437±0.005 | 0.523±0.007 |
|  | ✓ | 0.173±0.007 | 0.049±0.003 | 0.275±0.005 | 0.251±0.001 | 0.326±0.003 | 0.238±0.003 | 0.416±0.005 | 0.497±0.003 |
|  | ↓ | 12.0% | 21.3% | 3.0% | 4.8% | 2.8% | 2.2% | 4.6% | 5.0% |
| Full | - | 0.110±0.001 | 0.023±0.000 | 0.197±0.001 | 0.131±0.001 | 0.312±0.002 | 0.223±0.002 | 0.406±0.003 | 0.492±0.004 |

|  | CondTSF | ETTm1 | | ETTm2 | | ETTh1 | | ETTh2 | |
|---|---|---|---|---|---|---|---|---|---|
|  |  | MAE | MSE | MAE | MSE | MAE | MSE | MAE | MSE |
| Random | - | 0.693±0.041 | 0.913±0.095 | 0.629±0.065 | 0.724±0.155 | 0.742±0.055 | 1.027±0.129 | 0.691±0.140 | 0.887±0.294 |
| DC | ✗ | 0.603±0.045 | 0.730±0.075 | 0.490±0.018 | 0.410±0.032 | 0.724±0.007 | 0.977±0.022 | 0.634±0.054 | 0.711±0.115 |
|  | ✓ | 0.590±0.009 | 0.713±0.025 | 0.417±0.093 | 0.312±0.116 | 0.704±0.002 | 0.915±0.006 | 0.566±0.008 | 0.562±0.018 |
|  | ↓ | 2.2% | 2.4% | 14.8% | 24.0% | 2.8% | 6.3% | 10.7% | 21.0% |
| MTT | ✗ | 0.520±0.022 | 0.522±0.035 | 0.285±0.010 | 0.143±0.008 | 0.480±0.009 | 0.467±0.017 | 0.329±0.009 | 0.199±0.008 |
|  | ✓ | 0.462±0.006 | 0.476±0.012 | 0.265±0.009 | 0.130±0.007 | 0.428±0.009 | 0.383±0.012 | 0.303±0.007 | 0.177±0.008 |
|  | ↓ | 11.1% | 8.8% | 6.9% | 9.1% | 10.9% | 18.0% | 7.8% | 11.0% |
| PP | ✗ | 0.538±0.041 | 0.558±0.062 | 0.285±0.008 | 0.144±0.007 | 0.477±0.006 | 0.462±0.012 | 0.330±0.004 | 0.201±0.004 |
|  | ✓ | 0.466±0.006 | 0.485±0.022 | 0.271±0.009 | 0.135±0.008 | 0.442±0.016 | 0.405±0.029 | 0.323±0.004 | 0.198±0.005 |
|  | ↓ | 13.3% | 13.2% | 5.2% | 5.7% | 7.3% | 12.3% | 2.4% | 1.1% |
| TESLA | ✗ | 0.519±0.014 | 0.558±0.050 | 0.295±0.005 | 0.155±0.004 | 0.542±0.015 | 0.603±0.037 | 0.339±0.006 | 0.213±0.005 |
|  | ✓ | 0.480±0.023 | 0.507±0.061 | 0.288±0.004 | 0.152±0.004 | 0.480±0.010 | 0.471±0.020 | 0.327±0.005 | 0.205±0.004 |
|  | ↓ | 7.5% | 8.2% | 2.5% | 1.7% | 11.5% | 21.9% | 3.3% | 3.8% |
| FTD | ✗ | 0.531±0.023 | 0.539±0.036 | 0.293±0.016 | 0.150±0.014 | 0.480±0.009 | 0.464±0.018 | 0.327±0.010 | 0.197±0.009 |
|  | ✓ | 0.469±0.005 | 0.493±0.018 | 0.264±0.003 | 0.130±0.002 | 0.440±0.006 | 0.400±0.012 | 0.308±0.008 | 0.181±0.009 |
|  | ↓ | 11.7% | 8.5% | 9.7% | 13.3% | 8.3% | 13.7% | 5.8% | 7.8% |
| DATM | ✗ | 0.497±0.013 | 0.513±0.011 | 0.285±0.006 | 0.144±0.005 | 0.480±0.012 | 0.464±0.026 | 0.327±0.005 | 0.196±0.005 |
|  | ✓ | 0.493±0.006 | 0.495±0.009 | 0.268±0.008 | 0.131±0.007 | 0.429±0.033 | 0.385±0.053 | 0.299±0.007 | 0.172±0.007 |
|  | ↓ | 0.9% | 3.4% | 5.7% | 8.6% | 10.6% | 17.1% | 8.7% | 12.1% |
| Full | - | 0.432±0.001 | 0.473±0.001 | 0.230±0.001 | 0.113±0.001 | 0.389±0.003 | 0.339±0.004 | 0.276±0.002 | 0.166±0.002 |

We observe that CondTSF consistently improves the performance of backbone models with all condensing ratios, suggesting the effectiveness of CondTSF with different condensing ratios.

# C  Performance of CondTSF with Non-parameter-matching Based Methods

We distill the dataset using the standard condensing ratio. The information on condensation is shown in Table.4. We conduct experiments on CondTSF with non-parameter-matching based methods. We use DM[50], IDM[52], KIP[33], FRePo[55] as backbone methods. The performance is shown in Table.8.

Results show that using CondTSF to optimize only one of the two optimizable terms can also boost the performance.

Table 8: Distill performance of different dataset condensation methods. For each method, ✗means CondTSF is not used, ✓means CondTSF is used, and ↓ means the decreased percentage of test error after CondTSF is applied. Five synthetic datasets are distilled and the average and standard deviation are reported.

| | CondTSF | ExchangeRate | | Weather | | Electricity | | Traffic | |
|---|---|---|---|---|---|---|---|---|---|
| | | MAE | MSE | MAE | MSE | MAE | MSE | MAE | MSE |
| Random | - | 0.730±0.168 | 0.957±0.397 | 0.566±0.056 | 0.708±0.135 | 0.832±0.024 | 1.080±0.058 | 0.845±0.007 | 1.343±0.017 |
| DM | ✗ | 0.772±0.016 | 0.990±0.061 | 0.483±0.063 | 0.540±0.128 | 0.818±0.011 | 1.048±0.034 | 0.830±0.016 | 1.299±0.048 |
| | ✓ | 0.697±0.030 | 0.832±0.072 | 0.477±0.047 | 0.513±0.082 | 0.817±0.011 | 1.043±0.030 | 0.812±0.013 | 1.253±0.035 |
| | ↓ | 9.6% | 16.0% | 1.2% | 5.0% | 0.1% | 0.4% | 2.3% | 3.5% |
| IDM | ✗ | 0.708±0.107 | 0.871±0.257 | 0.517±0.052 | 0.594±0.105 | 0.836±0.012 | 1.087±0.032 | 0.823±0.022 | 1.287±0.055 |
| | ✓ | 0.683±0.120 | 0.805±0.247 | 0.504±0.055 | 0.570±0.116 | 0.819±0.023 | 1.050±0.055 | 0.804±0.020 | 1.231±0.055 |
| | ↓ | 3.5% | 7.5% | 2.4% | 4.0% | 2.0% | 3.4% | 2.4% | 4.4% |
| KIP | ✗ | 0.538±0.026 | 0.467±0.032 | 0.316±0.016 | 0.297±0.008 | 0.817±0.010 | 1.040±0.032 | 0.834±0.006 | 1.314±0.027 |
| | ✓ | 0.217±0.009 | 0.079±0.007 | 0.313±0.007 | 0.297±0.003 | 0.812±0.021 | 1.037±0.044 | 0.830±0.011 | 1.278±0.034 |
| | ↓ | 59.6% | 83.2% | 1.0% | 0.0% | 0.6% | 0.3% | 0.4% | 2.8% |
| FRePo | ✗ | 0.518±0.030 | 0.471±0.045 | 0.424±0.023 | 0.403±0.033 | 0.590±0.023 | 0.554±0.037 | 0.615±0.015 | 0.789±0.037 |
| | ✓ | 0.270±0.021 | 0.122±0.021 | 0.330±0.031 | 0.288±0.031 | 0.464±0.011 | 0.373±0.010 | 0.518±0.011 | 0.601±0.021 |
| | ↓ | 47.8% | 74.1% | 22.1% | 28.4% | 21.2% | 32.6% | 15.8% | 23.8% |
| Full | - | 0.110±0.001 | 0.023±0.000 | 0.197±0.001 | 0.131±0.001 | 0.312±0.002 | 0.223±0.002 | 0.406±0.003 | 0.492±0.004 |

| | CondTSF | ETTm1 | | ETTm2 | | ETTh1 | | ETTh2 | |
|---|---|---|---|---|---|---|---|---|---|
| | | MAE | MSE | MAE | MSE | MAE | MSE | MAE | MSE |
| Random | - | 0.697±0.054 | 0.934±0.105 | 0.629±0.129 | 0.747±0.285 | 0.725±0.067 | 0.995±0.152 | 0.645±0.118 | 0.763±0.251 |
| DM | ✗ | 0.684±0.063 | 0.903±0.129 | 0.641±0.129 | 0.782±0.254 | 0.722±0.040 | 0.977±0.094 | 0.703±0.079 | 0.895±0.189 |
| | ✓ | 0.651±0.041 | 0.826±0.089 | 0.614±0.130 | 0.706±0.322 | 0.713±0.035 | 0.950±0.077 | 0.615±0.133 | 0.706±0.267 |
| | ↓ | 4.8% | 8.5% | 4.3% | 9.7% | 1.2% | 2.8% | 12.5% | 21.1% |
| IDM | ✗ | 0.657±0.047 | 0.841±0.094 | 0.648±0.155 | 0.811±0.297 | 0.713±0.055 | 0.956±0.124 | 0.667±0.121 | 0.823±0.252 |
| | ✓ | 0.648±0.025 | 0.816±0.040 | 0.610±0.131 | 0.698±0.255 | 0.694±0.039 | 0.912±0.080 | 0.573±0.161 | 0.632±0.314 |
| | ↓ | 1.4% | 3.0% | 5.8% | 13.9% | 2.6% | 4.7% | 14.1% | 23.2% |
| KIP | ✗ | 0.581±0.002 | 0.736±0.012 | 0.316±0.002 | 0.171±0.002 | 0.685±0.021 | 0.861±0.028 | 0.576±0.114 | 0.575±0.198 |
| | ✓ | 0.581±0.001 | 0.723±0.014 | 0.290±0.002 | 0.151±0.002 | 0.602±0.036 | 0.709±0.082 | 0.400±0.054 | 0.282±0.061 |
| | ↓ | 0.0% | 1.8% | 8.0% | 11.7% | 12.1% | 17.6% | 30.6% | 51.0% |
| FRePo | ✗ | 0.596±0.015 | 0.670±0.040 | 0.572±0.023 | 0.556±0.053 | 0.640±0.014 | 0.759±0.024 | 0.549±0.077 | 0.528±0.131 |
| | ✓ | 0.551±0.011 | 0.581±0.016 | 0.424±0.024 | 0.303±0.039 | 0.566±0.005 | 0.617±0.008 | 0.430±0.064 | 0.325±0.079 |
| | ↓ | 7.6% | 13.2% | 25.8% | 45.5% | 11.6% | 18.7% | 21.6% | 38.4% |
| Full | - | 0.432±0.001 | 0.473±0.001 | 0.230±0.001 | 0.113±0.001 | 0.389±0.003 | 0.339±0.004 | 0.276±0.002 | 0.166±0.002 |

# D  Performance of CondTSF with Non-Linear Expert Models

We distill the dataset using the standard condensing ratio. The information on condensation is shown in Table.4. We conduct experiments on distilling dataset with non-linear expert models. We use MTT[3], TESLA[5], and DATM[11] as backbone methods. The performance of using a **CNN** as the expert model is shown in Table.9, and the performance of using a **3-layer-MLP** as the expert model is shown in Table.10.

Results show that CondTSF is also effective with non-linear expert models.

Table 9: Distill performance of different dataset condensation methods with **CNN** as the expert model. For each method, ✗means CondTSF is not used, ✓means CondTSF is used, and ↓ means the decreased percentage of test error after CondTSF is applied. Five synthetic datasets are distilled and the average and standard deviation are reported.

| | CondTSF | ExchangeRate | | Weather | | Electricity | | Traffic | |
|---|---|---|---|---|---|---|---|---|---|
| | | MAE | MSE | MAE | MSE | MAE | MSE | MAE | MSE |
| Random | - | 0.830±0.059 | 1.002±0.127 | 0.504±0.018 | 0.526±0.030 | 0.762±0.016 | 0.882±0.034 | 0.732±0.023 | 1.041±0.053 |
| MTT | ✗ | 0.372±0.028 | 0.237±0.031 | 0.314±0.013 | 0.278±0.007 | 0.482±0.009 | 0.393±0.012 | 0.662±0.011 | 0.906±0.026 |
| | ✓ | 0.140±0.011 | 0.063±0.003 | 0.246±0.008 | 0.120±0.004 | 0.357±0.005 | 0.267±0.005 | 0.451±0.013 | 0.519±0.009 |
| | ↓ | 62.4% | 73.4% | 21.7% | 56.8% | 25.9% | 32.1% | 31.9% | 42.7% |
| TESLA | ✗ | 0.378±0.007 | 0.242±0.007 | 0.310±0.015 | 0.292±0.012 | 0.516±0.005 | 0.430±0.009 | 0.655±0.020 | 0.900±0.045 |
| | ✓ | 0.134±0.012 | 0.058±0.002 | 0.253±0.007 | 0.137±0.002 | 0.374±0.008 | 0.267±0.007 | 0.528±0.011 | 0.632±0.020 |
| | ↓ | 64.6% | 76.0% | 18.4% | 53.1% | 27.5% | 37.9% | 19.4% | 29.8% |
| DATM | ✗ | 0.331±0.011 | 0.179±0.013 | 0.335±0.009 | 0.294±0.008 | 0.504±0.008 | 0.432±0.009 | 0.587±0.009 | 0.742±0.014 |
| | ✓ | 0.137±0.017 | 0.059±0.004 | 0.291±0.005 | 0.261±0.004 | 0.355±0.006 | 0.252±0.006 | 0.452±0.009 | 0.526±0.011 |
| | ↓ | 58.6% | 67.0% | 13.1% | 11.2% | 29.6% | 41.7% | 23.0% | 29.1% |

| | CondTSF | ETTm1 | | ETTm2 | | ETTh1 | | ETTh2 | |
|---|---|---|---|---|---|---|---|---|---|
| | | MAE | MSE | MAE | MSE | MAE | MSE | MAE | MSE |
| Random | - | 0.691±0.012 | 0.856±0.030 | 0.723±0.027 | 0.846±0.052 | 0.727±0.015 | 0.931±0.030 | 0.722±0.015 | 0.847±0.033 |
| MTT | ✗ | 0.550±0.006 | 0.585±0.011 | 0.347±0.008 | 0.205±0.009 | 0.644±0.013 | 0.788±0.036 | 0.371±0.016 | 0.245±0.020 |
| | ✓ | 0.482±0.007 | 0.507±0.007 | 0.236±0.009 | 0.111±0.005 | 0.460±0.008 | 0.437±0.014 | 0.297±0.011 | 0.173±0.009 |
| | ↓ | 12.4% | 13.3% | 32.0% | 45.9% | 28.6% | 44.5% | 19.9% | 29.4% |
| TESLA | ✗ | 0.544±0.003 | 0.583±0.007 | 0.359±0.013 | 0.222±0.016 | 0.634±0.010 | 0.757±0.037 | 0.365±0.008 | 0.240±0.009 |
| | ✓ | 0.499±0.003 | 0.492±0.007 | 0.251±0.009 | 0.119±0.005 | 0.473±0.009 | 0.458±0.018 | 0.293±0.004 | 0.170±0.002 |
| | ↓ | 8.3% | 15.6% | 30.1% | 46.4% | 25.4% | 39.5% | 19.7% | 29.2% |
| DATM | ✗ | 0.566±0.008 | 0.598±0.011 | 0.318±0.013 | 0.174±0.014 | 0.633±0.009 | 0.773±0.025 | 0.349±0.012 | 0.221±0.012 |
| | ✓ | 0.518±0.007 | 0.521±0.012 | 0.231±0.002 | 0.107±0.001 | 0.451±0.005 | 0.418±0.007 | 0.290±0.007 | 0.165±0.005 |
| | ↓ | 8.5% | 12.9% | 27.4% | 38.5% | 28.8% | 45.9% | 16.9% | 25.3% |

Table 10: Distill performance of different dataset condensation methods with **3-layer-MLP** as the expert model. For each method, ✗means CondTSF is not used, ✓means CondTSF is used, and ↓ means the decreased percentage of test error after CondTSF is applied. Five synthetic datasets are distilled and the average and standard deviation are reported.

| | CondTSF | ExchangeRate | | Weather | | Electricity | | Traffic | |
|---|---|---|---|---|---|---|---|---|---|
| | | MAE | MSE | MAE | MSE | MAE | MSE | MAE | MSE |
| Random | - | 0.932±0.058 | 1.243±0.144 | 0.562±0.013 | 0.642±0.030 | 0.796±0.013 | 0.955±0.036 | 0.751±0.012 | 1.112±0.025 |
| MTT | ✗ | 0.364±0.027 | 0.211±0.030 | 0.311±0.007 | 0.297±0.009 | 0.475±0.011 | 0.380±0.016 | 0.633±0.011 | 0.841±0.021 |
| | ✓ | 0.139±0.004 | 0.034±0.002 | 0.248±0.015 | 0.135±0.006 | 0.375±0.007 | 0.268±0.008 | 0.501±0.008 | 0.589±0.016 |
| | ↓ | 61.8% | 83.9% | 20.3% | 54.5% | 21.1% | 29.5% | 20.9% | 30.0% |
| TESLA | ✗ | 0.352±0.012 | 0.209±0.010 | 0.297±0.001 | 0.272±0.003 | 0.525±0.004 | 0.462±0.008 | 0.594±0.009 | 0.751±0.022 |
| | ✓ | 0.128±0.017 | 0.027±0.007 | 0.252±0.006 | 0.135±0.001 | 0.397±0.012 | 0.297±0.014 | 0.488±0.006 | 0.577±0.009 |
| | ↓ | 63.6% | 87.1% | 15.2% | 50.4% | 24.4% | 35.7% | 17.8% | 23.2% |
| DATM | ✗ | 0.326±0.016 | 0.177±0.016 | 0.349±0.002 | 0.301±0.001 | 0.517±0.008 | 0.441±0.011 | 0.622±0.007 | 0.785±0.005 |
| | ✓ | 0.141±0.005 | 0.042±0.001 | 0.254±0.010 | 0.126±0.004 | 0.385±0.009 | 0.280±0.008 | 0.496±0.006 | 0.582±0.011 |
| | ↓ | 56.7% | 76.3% | 27.2% | 58.1% | 25.5% | 36.5% | 20.3% | 25.9% |

| | CondTSF | ETTm1 | | ETTm2 | | ETTh1 | | ETTh2 | |
|---|---|---|---|---|---|---|---|---|---|
| | | MAE | MSE | MAE | MSE | MAE | MSE | MAE | MSE |
| Random | - | 0.692±0.013 | 0.855±0.027 | 0.762±0.012 | 0.955±0.037 | 0.687±0.010 | 0.838±0.022 | 0.763±0.019 | 0.945±0.050 |
| MTT | ✗ | 0.564±0.012 | 0.655±0.022 | 0.362±0.011 | 0.219±0.012 | 0.615±0.002 | 0.732±0.015 | 0.354±0.011 | 0.228±0.012 |
| | ✓ | 0.493±0.007 | 0.511±0.007 | 0.245±0.022 | 0.098±0.015 | 0.423±0.006 | 0.369±0.008 | 0.285±0.009 | 0.145±0.005 |
| | ↓ | 12.6% | 22.0% | 32.3% | 55.3% | 31.2% | 49.6% | 19.5% | 36.4% |
| TESLA | ✗ | 0.541±0.002 | 0.570±0.004 | 0.341±0.014 | 0.197±0.013 | 0.606±0.007 | 0.745±0.022 | 0.337±0.008 | 0.210±0.006 |
| | ✓ | 0.487±0.003 | 0.493±0.004 | 0.250±0.009 | 0.119±0.005 | 0.438±0.011 | 0.398±0.013 | 0.283±0.007 | 0.145±0.006 |
| | ↓ | 10.0% | 13.5% | 26.7% | 39.6% | 27.7% | 46.6% | 16.0% | 31.0% |
| DATM | ✗ | 0.558±0.009 | 0.611±0.021 | 0.329±0.006 | 0.186±0.005 | 0.593±0.022 | 0.741±0.063 | 0.350±0.009 | 0.222±0.008 |
| | ✓ | 0.498±0.006 | 0.482±0.006 | 0.234±0.005 | 0.107±0.001 | 0.437±0.006 | 0.397±0.005 | 0.286±0.005 | 0.167±0.004 |
| | ↓ | 10.8% | 21.1% | 28.9% | 42.5% | 26.3% | 46.4% | 18.3% | 24.8% |

# E  Performance Comparison of CondTSF and Smoothing

We distill the dataset using the one-shot condensing ratio. The information on condensation is shown in Table.2. We conduct experiments on distilling dataset with CondTSF and a low pass filter respectively. We use DLinear[45] as the expert model. The result is shown in Table.11.

Results indicate that using CondTSF is significantly more effective than using a low pass filter to smooth the distilled data.

Table 11: Distill performance of different dataset condensation methods. Five synthetic datasets are distilled and the average and standard deviation are reported.

| | ExchangeRate | | Weather | | Electricity | | Traffic | |
| --- | --- | --- | --- | --- | --- | --- | --- | --- |
| | MAE | MSE | MAE | MSE | MAE | MSE | MAE | MSE |
| Random | 0.783±0.090 | 1.070±0.246 | 0.530±0.084 | 0.647±0.159 | 0.840±0.017 | 1.102±0.031 | 0.854±0.018 | 1.350±0.043 |
| MTT | 0.778±0.084 | 0.964±0.136 | 0.509±0.065 | 0.538±0.085 | 0.747±0.012 | 0.840±0.019 | 0.742±0.010 | 1.052±0.024 |
| Smooth | 0.867±0.106 | 1.358±0.321 | 0.602±0.018 | 0.772±0.032 | 0.831±0.042 | 1.068±0.109 | 0.786±0.037 | 1.208±0.101 |
| MTT+Smooth | 0.620±0.024 | 0.636±0.062 | 0.501±0.012 | 0.527±0.028 | 0.788±0.027 | 0.963±0.061 | 0.836±0.022 | 1.291±0.057 |
| MTT+CondTSF | **0.195±0.007** | **0.061±0.004** | **0.326±0.009** | **0.284±0.007** | **0.391±0.003** | **0.284±0.004** | **0.494±0.022** | **0.579±0.037** |

| | ETTm1 | | ETTm2 | | ETTh1 | | ETTh2 | |
| --- | --- | --- | --- | --- | --- | --- | --- | --- |
| | MAE | MSE | MAE | MSE | MAE | MSE | MAE | MSE |
| Random | 0.728±0.033 | 0.993±0.082 | 0.695±0.011 | 0.889±0.030 | 0.756±0.035 | 1.059±0.083 | 0.749±0.037 | 1.013±0.089 |
| MTT | 0.653±0.019 | 0.771±0.040 | 0.685±0.022 | 0.754±0.051 | 0.693±0.009 | 0.845±0.023 | 0.719±0.006 | 0.827±0.016 |
| Smooth | 0.728±0.013 | 0.991±0.038 | 0.706±0.053 | 0.910±0.136 | 0.701±0.007 | 0.879±0.026 | 0.744±0.067 | 1.023±0.145 |
| MTT+Smooth | 0.656±0.014 | 0.919±0.040 | 0.619±0.029 | 0.684±0.062 | 0.680±0.012 | 0.874±0.044 | 0.704±0.053 | 0.910±0.136 |
| MTT+CondTSF | **0.491±0.004** | **0.502±0.008** | **0.347±0.028** | **0.202±0.028** | **0.532±0.014** | **0.580±0.029** | **0.329±0.003** | **0.205±0.002** |

# F  Ablation Study of CondTSF

We compare the changing trajectories of test error during the dataset condensation process. Since MTT has been proven to be a suitable backbone for CondTSF, we conduct experiments on different methods of plugging CondTSF into MTT. We utilize the standard condensing ratio as shown in Table.4.

Test error is calculated as such. After the synthetic data has been distilled, it is used to train 5 randomly initialized testing models. After training with the synthetic dataset, the models are tested on the test set sampled from the source dataset. MAE error is reported in the figures below.

## F.1  Performance of CondTSF with Different Gap

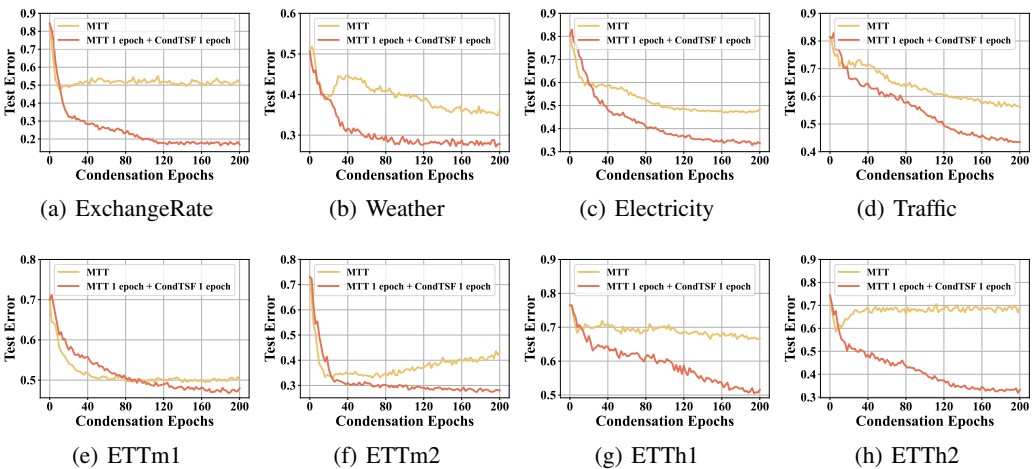

Figure 5: **Yellow:** Use MTT to distill for 200 epochs. **Orange:** Use MTT to distill for 200 epochs and use CondTSF to update in every epoch.

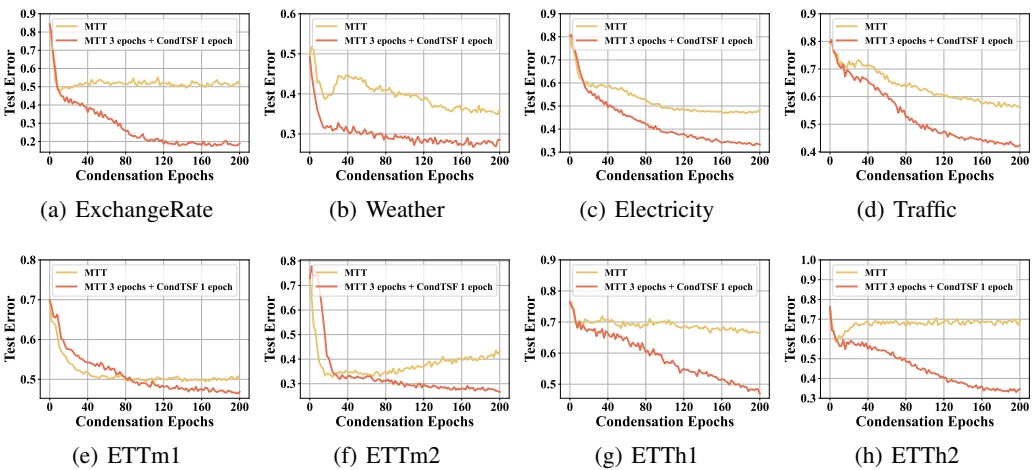

Figure 6: **Yellow:** Use MTT to distill for 200 epochs. **Orange:** Use MTT to distill for 200 epochs and use CondTSF to update every 3 epochs.

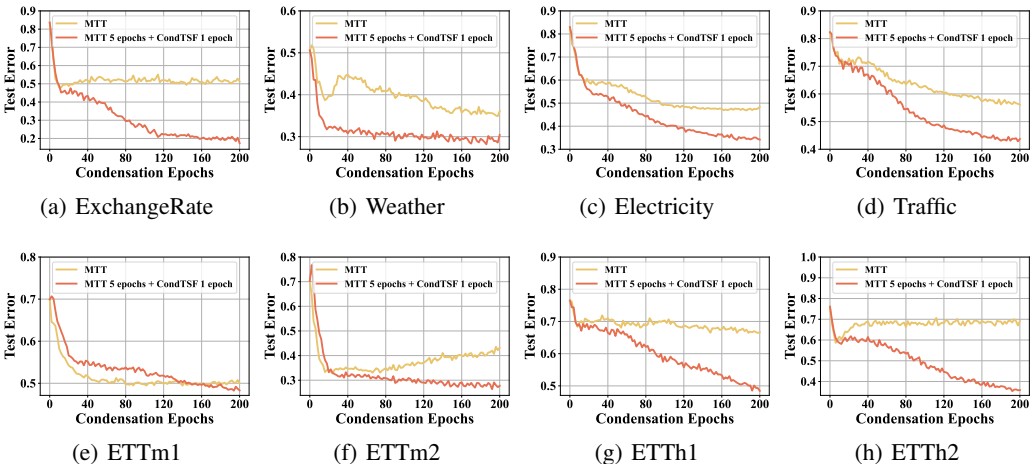

Figure 7: **Yellow:** Use MTT to distill for 200 epochs. **Orange:** Use MTT to distill for 200 epochs and use CondTSF to update every 5 epochs.

We observe that CondTSF consistently reduces the testing error with different utilization gaps.

## F.2 Relationship of Performance and Label Error

We also conduct experiments on label error and test error. We visualize the trajectory of label error $\mathcal{L}_{label}$ and test error through the distillation process. The results are shown in Fig.8.

- **Model 1:** Use MTT to distill for 200 epochs.
- **Model 2:** Use MTT to distill for 160 epochs and then use CondTSF to update for 40 epochs.

As shown in Fig.8, it can be observed that using MTT[3] leads to an increase in label error $\mathcal{L}_{label}$. While applying CondTSF effectively lowers the label error in the last 40 epochs, and therefore enhancing the performance.

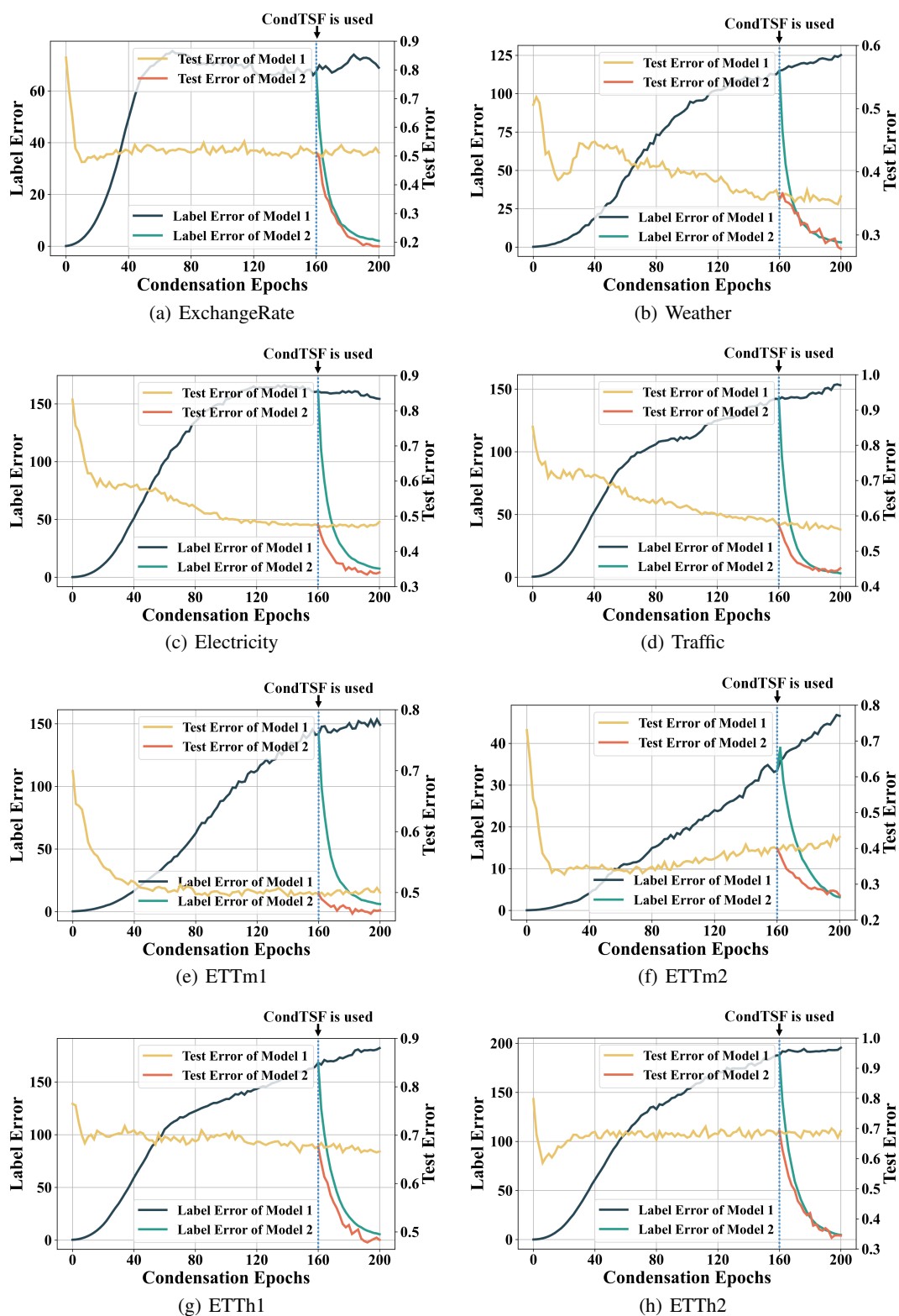

Figure 8: Visulization of the training curve of label error and test error during the distillation process.

# G    Parameter Sensitivity of CondTSF

We test CondTSF with different update gaps $G$ and additive update ratios $\beta$. We utilize the standard distill ratio as shown in Table.4. Our observations indicate that CondTSF displays a notable degree of robustness concerning these parameters. Specifically, the effectiveness of CondTSF persists when the update gap $G$ is moderately sized and additive update ratio $\beta$ is not excessively small.

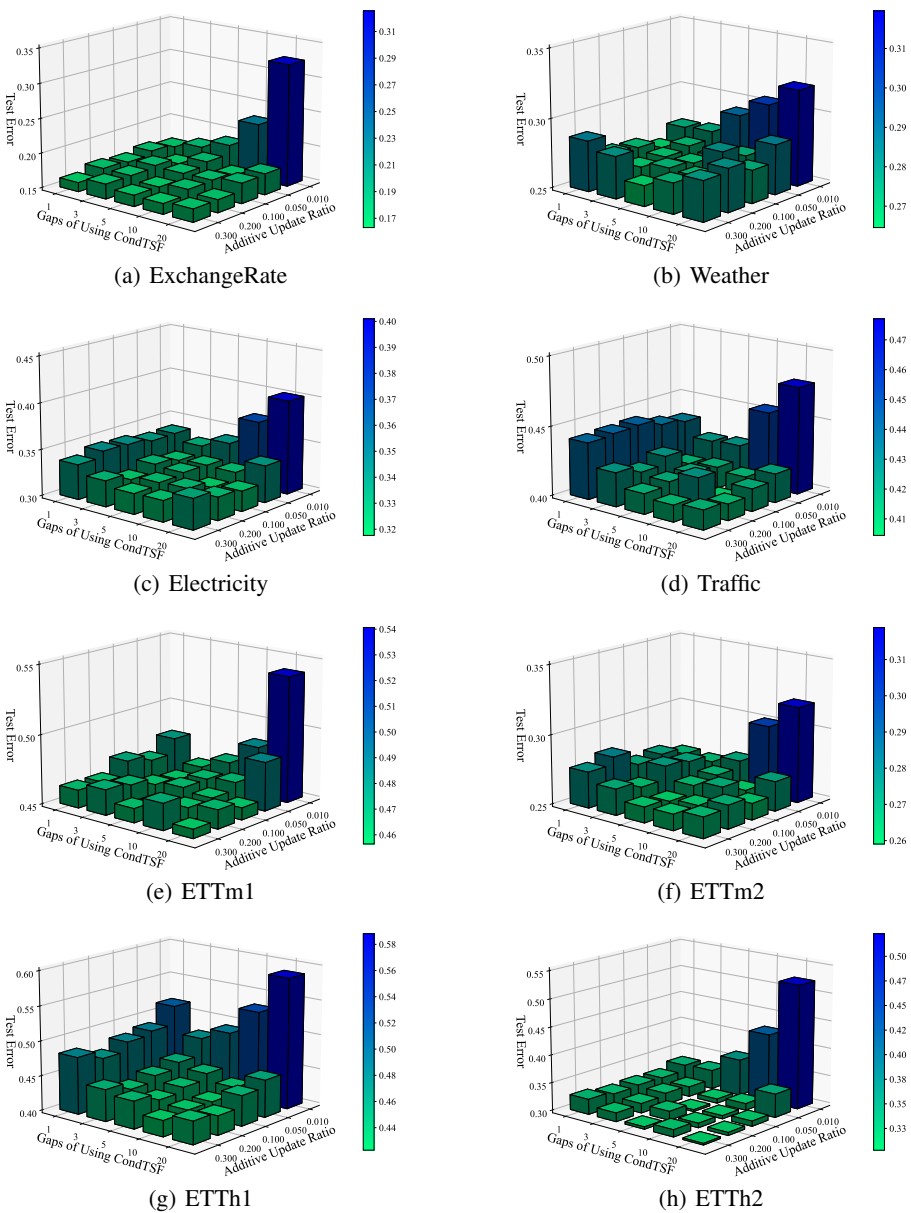

Figure 9: Performance of CondTSF with different update gaps and update ratios.

# H Visualization of Synthetic Data

We provide some visualization of synthetic data distilled by MTT[3] and MTT+CondTSF on all datasets. It is observed that the synthetic dataset distilled with CondTSF is smoother than the ones without CondTSF. Smoother data indicates more generalized features and therefore helps boost the performance.

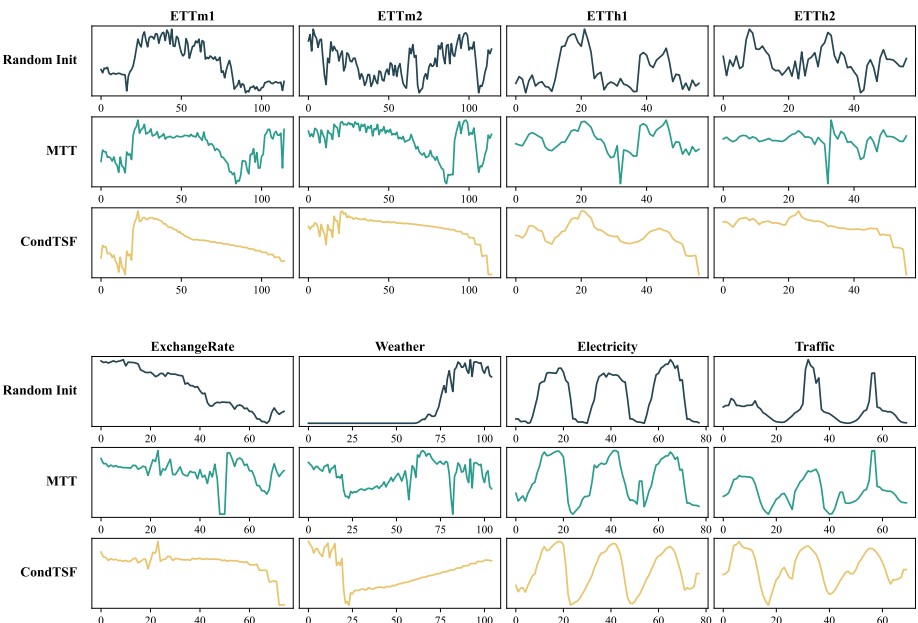

Figure 10: Visualization of synthetic data.

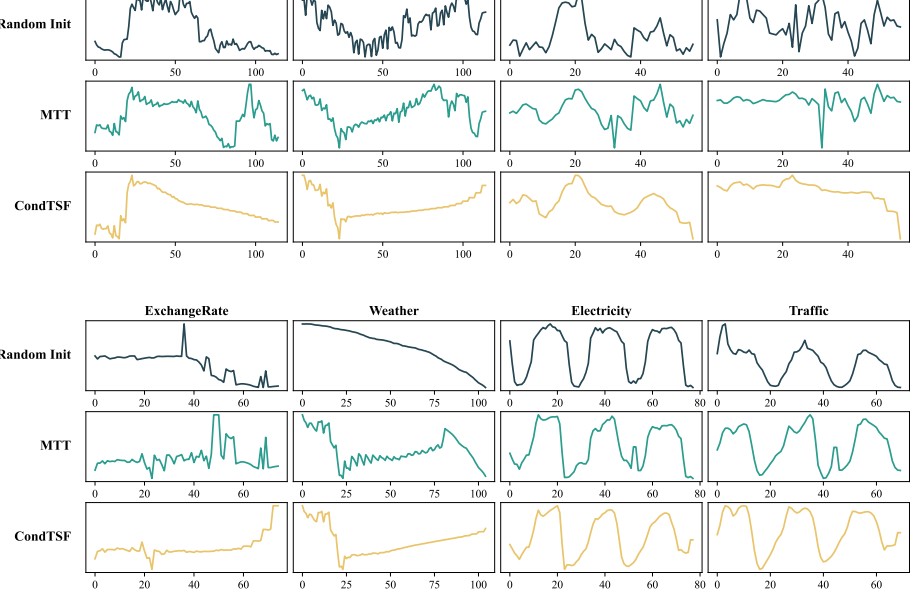

Figure 11: Visualization of synthetic data.

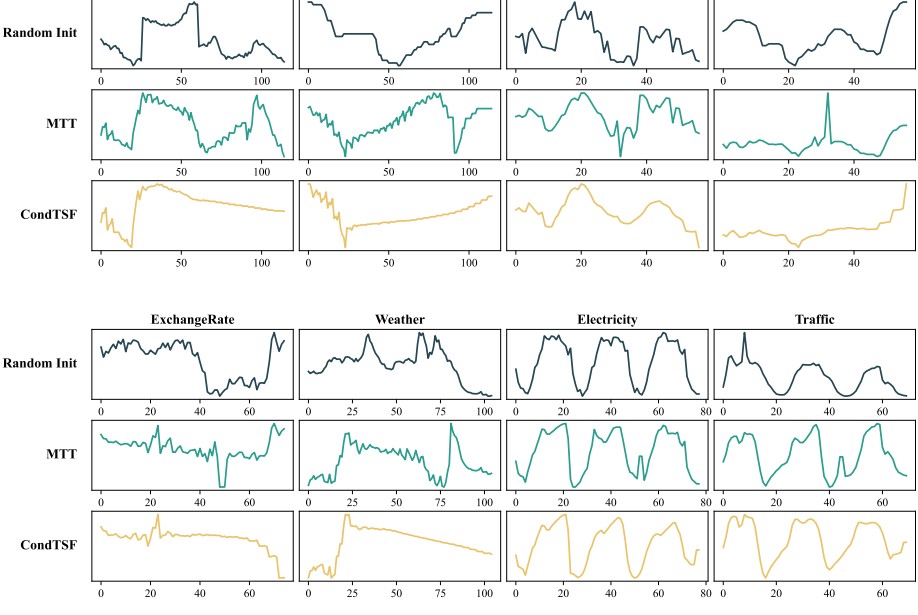

Figure 12: Visualization of synthetic data.

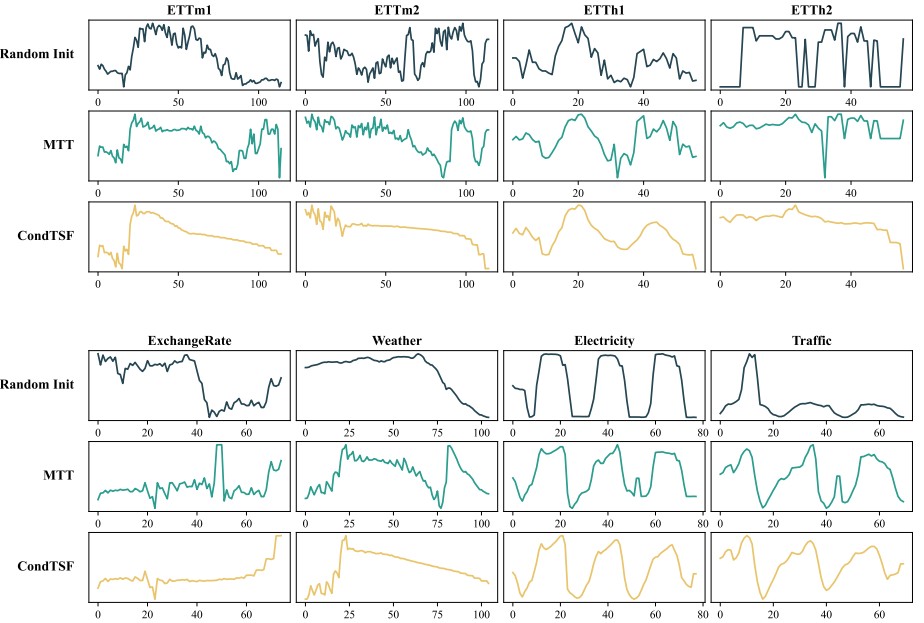

Figure 13: Visualization of synthetic data.

