# OpenReview forum: "CondTSF: One-line Plugin of Dataset Condensation for Time Series Forecasting"
_NeurIPS.cc/2024/Conference — NeurIPS 2024 poster_

### Official Review · Reviewer_s2gY · 2024-07-05

**Soundness:** 2
**Presentation:** 3
**Contribution:** 2
**Rating:** 5
**Confidence:** 3

**Summary:**

In this paper, the authors propose a dataset distillation approach by decomposing the distillation loss into two terms, the value term and the gradient term. Then, they derive bounds for these two terms that enable more effective dataset distillation for time-series forecasting. Strong performance is demonstrated in the conducted experimental evaluation

**Strengths:**

+ The paper is well-written and easy to follow.

+ Authors attempt to support their methodology through theoretical explanations

+ Strong performance is obtained in the experimental evaluation.

**Weaknesses:**

- p4. l121. The authors do not explain why this non-optimizable $\epsilon$ error occurs.

- l. 137 in principle we could directly optimize this gradient term without deriving any bound. Unless I am missing something, this is supported by DL frameworks (e.g., PyTorch). On the other hand, I understand that there might be computational benefits from deriving a bound that does not involve the gradients. However, authors should justify their choice (why we need to derive this bound) and if this is related to computational efficiency. Ideally, experiments should be provided to support their choice.

- At various points there are implicit assumptions the effect of which is not appropriately discussed, e.g., Eq. (9) is essentially supported by an experimental result in a Figure and a verbal description. I do not (empirically) disagree with the conclusion, but I can see cases where this would not hold (e.g., in financial time-series forecasting, where using the same data we can have vastly different (non-linear) hypotheses). Whether this alters the conclusions/and or the method, is not appropriately discussed.

- The updating process in (12) essentially moves the targets towards the predictions of the model. This is a reasonable result that is well-known to work. However, the way this is applied is now very close to target smoothing/model distillation approaches that essentially do the same thing (also applied in the context of time series forecasting as far as I know). There is some literature on model distillation approaches (and/or self-distillation) in the context of time-series forecasting and/or regression, which essentially demonstrate that this kind of "smoothing" can indeed increase performance. Therefore, this raises some questions on whether the method should be compared experimentally (or at least theoretically) with such approaches.

- Even though some ablation studies were provided in the Appendix, I didn't manage to find perhaps the most important result: What is the impact of the gradient term and what is the impact of the value term? What would happen if we remove the L_param loss in l. 9 of Alg 1? What would happen if we removed the value term?

- Isn't the method also applicable in the case of regression tasks? If I am not missing anything, I think the same methodology without any change (apart from the input to the models) can be applied also to handle generic regression tasks that also expand over time-series forecasting.


Minor: l.147-152: I think $k$ is used before being introduced.

**Questions:**

Please respond to the comments provided in the drawbacks. Overall, strong performance is demonstrated, but there are several aspects that should be improved.

**Limitations:**

Yes.

---

> ### Author Rebuttal · Authors · 2024-08-06
>
> We apologize for the ambiguity.
>
> **W1:** What does non-optimizable term $\epsilon$ mean?
>
> **A1:**
> - Please refer to Eq(4), since test label $x_{t+m:t+m+n}$ is unavailable during dataset condensation, we transform it to the prediction of an expert model $M_f$ on test data $x_{t:t+m}$ for further analysis. Since there is always an error between the prediction of the model and the ground truth, we use $\epsilon$ to denote it.
> - $\epsilon$ is optimized during the training process of expert model $M_f$. However, since we can only have access to the trained expert model $M_f$ during dataset condensation, $\epsilon$ is no longer optimizable.
>
> ---
>
> **W2:** Why derive a bound of the gradient term?
>
> **A2:**
> - Indeed, the original form of **gradient term** in Eq(5) is not directly optimizable. Test data $x_{t:t+m}$ exists in **gradient term**, which is unavailable during dataset condensation.
> - To solve this problem, please refer to Eq(6), we use Cauchy-Schwarz Inequality to decouple **gradient term**. Then we get the optimizable part $||\theta_f-\theta_s||^2$(the gradient of linear model is the parameter), and the non-optimizable part $||x_{t:t+m}-s_{t':t'+m}||^2$.
> - Thus, it is necessary to derive the upper bound of **gradient term** because the original one is not optimizable.
>
> ---
>
> **W3:** How to justify the empirical finding?
>
> **A3:** In the cases you mentioned, the model is theory-driven, which means the prediction of the model relies highly on the hypotheses and theories. However, in this case, the models are data-driven, and no additional hypotheses are made. The prediction of the model relies only on the training data and the model structure. Thus, in this case, if the models share the same architecture and are trained with the same dataset, empirically they output similar outputs.
>
> ---
>
> **W4:** How does our method compare to smoothing or model distillation?
>
> **A4:**
> - Firstly, model distillation and dataset condensation are different tasks. Model distillation updates student model parameter and produces distilled model. Dataset condensation updates the synthetic data and produces distilled dataset.
> - Secondly, model distillation aims at aligning the output of $M_s$ and $M_f$. The training of $M_s$ is directly supervised by $M_f$, where $L$ is the loss function, e.g. MSE.
>   $$M_s=\arg\min_{M} L(M(s),M_f(s))$$
>   However, in the case of our **value term** optimized by Eq(12), $M_s$ is trained alone using synthetic data $s$ and $M_f$ only has indirect access to the training of $M_s$.
>   $$s=\arg\min_s L(M_s(s),M_f(s))$$
>   $$\textbf{s.t.}\hspace{1em} M_s=\arg\min_M L_{train}(M, s)$$
>   Thus model distillation is not comparable to our method.
> - Meanwhile, we conduct experiments using LPF to perform target smoothing on the synthetic data and compare the results with CondTSF. Average MAE of 5 models is reported below. It shows that CondTSF is significantly more effective than target smoothing.
>
> ||Ex.Rate|Wea.|Elec.|Tra.|ETTm1|ETTm2|ETTh1|ETTh2|
> |-|-|-|-|-|-|-|-|-|
> |MTT|0.778|0.509|0.747|0.742|0.653|0.685|0.693|0.719|
> |Smooth|0.867|0.602|0.831|0.786|0.728|0.706|0.701|0.744|
> |MTT+Smooth|0.620|0.501|0.788|0.836|0.656|0.619|0.680|0.704|
> |MTT+CondTSF|0.195|0.326|0.391|0.494|0.491|0.347|0.532|0.329|
>
> ---
>
> **W5:** What if the value term or the gradient term is removed? What are the impacts of these two terms?
>
> **A5:** $L_{param}$ is the same form of Eq(8) in Sec.4.2, which is used to optimize **gradient term**. Please refer to l.143-145, our analysis shows that previous dataset condensation methods based on parameter matching, e.g. DC[1], MTT[2], etc. are optimizing the **gradient term**. Our method CondTSF is optimizing the **value term**.
> - If **value term** is removed: The method falls back to the previous dataset condensation methods based on parameter matching(DC, MTT, etc.), and the results can be found in Table.1,3,5,7,9. Moreover, please refer to Fig.8, in the beginning 160 epochs, only **gradient term** is optimized, in the last 40 epochs, **value term** is also optimized. Optimizing **value term** significantly enhances the performance.
> - If **gradient term** is removed: Please refer to l.222-224 and App.C, experiments on backbone methods that are not based on parameter matching, e.g. DM[3], IDM[4], etc. are also carried out. These backbone methods do not optimize the **gradient term**. Thus when combining these backbone methods with CondTSF, only **value term** is optimized, and the results are in Table.8. We conduct experiments that only optimize **value term** without any backbone methods, average MAE of 5 models is reported. The results are well-aligned with our analysis that optimizing **gradient term** or **value term** alone harms the performance.
>
> ||Ex.Rate|Wea.|Elec.|Tra.|ETTm1|ETTm2|ETTh1|ETTh2|
> |-|-|-|-|-|-|-|-|-|
> |MTT (only gradient term)|0.778|0.509|0.747|0.742|0.653|0.685|0.693|0.719|
> |CondTSF (only value term)|0.563|0.537|0.761|0.809|0.738|0.538|0.623|0.569|
> |MTT+CondTSF|0.195|0.326|0.391|0.494|0.491|0.347|0.532|0.329|
>
> [1] Dataset Condensation with Gradient Matching
>
> [2] Dataset Distillation by Matching Training Trajectories
>
> [3] Dataset Condensation with Distribution Matching
>
> [4] Improved Distribution Matching for Dataset Condensation
>
> ---
>
> **W6:** Is this method applicable to all regression tasks?
>
> **A6:** Our proposed CondTSF is applicable to generic regression. Meanwhile, we have designs that are related to time series forecasting. In time series forecasting, when generating training data, the data is usually sampled overlap from the dataset. For instance, the data points can be sampled as $x_{0:24}, x_{4:28}$, etc. Please refer to Eq(12), inspired by the overlap property, we utilize an additive method to gradually update the synthetic data instead of directly using $M_f(s_{t':t'+m})$ to overwrite $s_{t'+m:t'+m+n}$ to avoid vibrations.
>
> ---
>
> **Minor:** $k$ in l.147-152 used before introduced.
>
> **A:** We are sorry for the ambiguity. We will correct that in next version.

---

> > ### Comment · Reviewer_s2gY · 2024-08-11
> >
> > I would like to thank the authors for providing detailed responses to all of my questions, which were - to a significant degree - indeed addressed. Therefore, I am increasing my rating.

---

> > > ### Author Response · Authors · 2024-08-11
> > > **Thanks for the comments and raising the score**
> > >
> > > Dear reviewer s2gY,
> > >
> > > We want to thank you for the valuable advice and for raising the score of our paper.
> > >
> > > We will incorporate your suggestions, especially clarifying the terms, and adding the experiment results reported in the rebuttal.
> > >
> > > Best,
> > >
> > > Authors

---

### Official Review · Reviewer_DVXT · 2024-07-11

**Soundness:** 2
**Presentation:** 3
**Contribution:** 2
**Rating:** 4
**Confidence:** 3

**Summary:**

This paper explores dataset condensation, which generates a small dataset for training of deep neural networks, for time series forecasting. Specifically, it proposes a one-line dataset condensation plugin designed specifically for time series forecasting. It first formulates the optimization objective of dataset condensation for time series forecasting, and based on which a reformulation is proposed and further theoretical analysis is performed. Based on the proposed theoretical analysis, this paper proposes a simple data condensation plugin CondTSF. Experiments on 8 time series datasets are performed to demonstrate the effectiveness of the proposed CondTSF.

**Strengths:**

1. It proposes a simple dataset condensation plugin.
2. A theoretical analysis is provided.

**Weaknesses:**

1. Dataset condensation is a strategy specifically designed for training large dataset. However, the data size in time series is usually significantly smaller than that in NLP and CV. In my experience, the computational demand to deal with time series data may not be an urgent request in real-world time series applications. The topic discussed in this paper seems interesting given the current program of different foundation models, but I highly doubt it applicability in the field of time series.
2.  A related problem is that how the time series foundation model performs remain unclear. Exploring the foundation model for time series may be a promising direction, but it is not widely accepted and deployed in real-world applications. In fact, considering the semantics of time series is highly dependent on the underlying applications, so far I have not observed any foundation model can provide good zero-shot learning capability.

**Questions:**

Please refer to the weakness part.

**Limitations:**

Please refer to the weakness part.

---

> ### Author Rebuttal · Authors · 2024-08-06
>
> **W1:** The demand for dataset condensation for time series forecasting is not urgent. What are the real applications of dataset condensation for time series foundation models?
>
> **A1:**
> - Firstly, dataset condensation is also important in other aspects besides reducing computational cost. For instance, data condensation is proven to benefit downstream tasks such as continual learning[1], privacy[2], etc. These tasks are vital for some aspects of the time series, such as finances.
> - Secondly, in the era of large models, Neural Architecture Search(NAS) of large time series foundation models is still computationally intensive[3]. Utilizing condensed data to identify a smaller parameter set and conducting full training on this set can improve efficiency.
> - Moreover, our proposed method could serve as a prototype and shed light on the related research area.
>
> [1] An Efficient Dataset Condensation Plugin and Its Application to Continual Learning
>
> [2] Privacy for free: How does dataset condensation help privacy
>
> [3] Dataset Condensation for Time Series Classification via Dual Domain Matching
>
> ----
>
> **W2:** How the time series foundation model performs remains unclear, and most are not good at zero-shot learning.
>
> **A2:**
> - Indeed, nowadays foundation models do not behave well in time series forecasting[1].
> - However, more explorations of foundation models are carried out[2]. Therefore, our dataset condensation method can be used to lower the computational cost and accelerate the exploration of foundation models on time series tasks[3][4][5].
> - Meanwhile, our proposed method points out an interesting direction in time series forecasting tasks. More efforts can be devoted to this field and our method can serve as a prototype.
>
> [1] Are Language Models Actually Useful for Time Series Forecasting?
>
> [2] A Survey of Time Series Foundation Models: Generalizing Time Series Representation with Large Language Model
>
> [3] Time-llm: Time series forecasting by reprogramming large language models
>
> [4] Timegpt-1
>
> [5] A decoder-only foundation model for time-series forecasting

---

> > ### Comment · Reviewer_DVXT · 2024-08-11
> >
> > Thanks for the reviewer for the detailed response. I acknowledge I have read the rebuttal.

---

> > > ### Author Response · Authors · 2024-08-11
> > > **Thanks for the comments**
> > >
> > > Dear reviewer DVXT,
> > >
> > > We want to thank you for your valuable advice and we will clarify the motivation and the pragmatic usage of dataset condensation for time series in the paper.
> > >
> > > Meanwhile, we are eager for the opportunity to address any remaining concerns you might have before the discussion period concludes. We sincerely appreciate your and other reviewers' dedication and time invested in reviewing our work.
> > >
> > > Best,
> > >
> > > Authors

---

### Official Review · Reviewer_TXkn · 2024-07-12

**Soundness:** 3
**Presentation:** 1
**Contribution:** 3
**Rating:** 4
**Confidence:** 3

**Summary:**

In this paper the authors study the question of dataset condensation / distillation in the context of time series forecasting. They propose a decomposition that could be used to empirically estimate bound the point-wise forecast MSE loss. Based on this decomposition the authors proposed a condensation plug in on top of existing data condensation method that updates the distilled dataset with the estimated full model forecast. Through empirical study they show that with this plug in the resulting condensed dataset leads to better models across different model classes and benchmark forecasting tasks.

**Strengths:**

The paper attempts a new and simple solution to an important training data efficiency problem. It also attempts a novel prospect of analyzing the dataset distillation through the loss decomposition.

**Weaknesses:**

The major weakness is the writing. It is hard to follow in its current shape, in particular the math description of the research question should be more rigorous. To name a few:
- What are the sample space of x and s? They now look like a single univariate time series. It is the case?
- What is t' in Eqn (5)? Is it summed over? If so is it double sum within sum_t?
- How is Eqn (7) and (8) optimized (also in Algo 1).
Similarly regarding writing, the paper does not motivate the research question well, and has not provided enough background details on its base distillation methods (Line 200 Model Settings), especially MTT(3). As a result it is hard to evaluate the novelty of the paper when the main idea is not clearly delivered.

Though mentioned as a limitation, the assumption of linear model trivializes the theories, which ties the proposed method to DLinear expert, and makes it hard to understand how the learning from the synthetic data generalizes beyond the expert's model class. There are similar simplifications, e.g. the empirical approximation as in Eqn (9) that's observational.

**Questions:**

Besides the questions in the weakness:

- While the distill ratio in Table 2, e.g. condensing the datasets into one training example, can be used to show the extremity of the distillation process,  it is questionable what this one-shot learning means for the MLP, LSTM, CNN involved. Any insights, especially why to choose them over the few-shot study in, e.g. Appendix F (this one has visualization as well) to be in the main paper?
- As now the expert model is more actively involved through the two decomposition steps to touch the synthetic data, how does the capacity of the expert impact the fidelity of the synthetic dataset versus the full one? What about at least one non linear expert just to get some insights?
- The method proposed here does not utilize any time series concepts. If so this is an application on multivariate regression with linear expert. What's the novelty beyond that for time series exclusively?

**Limitations:**

The authors addressed the limitations.

---

> ### Author Rebuttal · Authors · 2024-08-06
>
> We apologize for the ambiguity.
>
> **W1:** What are the sample space of $x, s$? Are they univariate time series?
>
> **A1:** Yes, $f, x, s$ are univariate time series. Please refer to Sec.3, l.93-95. A time series dataset can be viewed as a long vector. We cut the dataset into two vectors and obtain train set $f$ and test set $x$. Both $x$ and $f$ are vectors. Then we sample a short continuous part of train set $f$ as the initial synthetic dataset $s$. $s$ is still a vector.
>
> ---
>
> **W2:** What is $t'$ in Eq(5)? Is it summed over?
>
> **A2:**
> - We use $s_{t':t'+m}$ to denote a vector(part of $s$) with length $m$ starting from position $t'$. $t'$ is arbitrarily chosen.
> - Please refer to Eq(16) and l.444-446, $s_{t':t'+m}$ is an arbitrary vector used to perform Taylor expansion to get the value of $M(x_{t:t+m})$. The idea throughout our paper is that test set $x$ is unavailable during dataset condensation, thus we need to substitute $x$ with accessible synthetic dataset $s$ so the optimization objective becomes optimizable.
> - The sum in Eq(5) is only performed on $t$, not $t'$. Please refer to Eq(3) and App.A1, $\sum_t$ is used in test error to sum over the test set $x$. For each $x_{t:t+m}$, we use an arbitrary $s_{t':t'+m}$ to perform Taylor Expansion, thus $t'$ is not summed over.
>
> ----
>
> **W3:** How are Eq(7) and(8) optimized? Lack of information on base methods and the novelty is not clear.
>
> **A3:**
> - Eq(7) and(8) are optimized the same way as MTT[1], which is a well-acknowledged dataset condensation method. According to our analysis, MTT is optimizing the **gradient term**. There are multiple dataset condensation methods that use MTT as the backbone, such as TESLA[2], FTD[3], DATM[4], etc. We will add introduction of MTT in appendix in the next version.
> - MTT and all other backbone methods are designed for classification. They perform poorly on time series forecasting. The novelty of our method is that CondTSF enhanced the performance of all previous methods on time series forecasting.
>
> [1] Dataset Distillation by Matching Training Trajectories
>
> [2] Scaling Up Dataset Distillation to ImageNet-1K with Constant Memory
>
> [3] Minimizing the Accumulated Trajectory Error to Improve Dataset Distillation
>
> [4] Towards Lossless Dataset Distillation via Difficulty-Aligned Trajectory Matching
>
> ---
>
> **W4.1:** The hypothesis of linear model trivialized the work.
>
> **A4:**
> - Please refer to App.A1 and Eq(16), using nonlinear models will add 2nd and higher order terms to the Taylor Expansion. We ignore higher order terms because their effect on the output is minor. Meanwhile, although linear models are simple in structure, they perform well on time series forecasting and outperform complicated transformer models[1].
> - We add experiments on using nonlinear models as expert models, please refer to **A7**.
>
> [1] Are Transformers Effective for Time Series Forecasting?
>
> ---
>
> **W4.2:** The observation in Eq(9) is a simplification.
>
> **A5:** The observation in Eq(9) is not a simplification. Please refer to l.157-159, The test model parameter is unknown during dataset condensation, thus we need to substitute them with accessible parameters. Thus Eq(9) is necessary, otherwise the **value term** is not optimizable.
>
> ---
>
> **Q1:** Why is the result of one-shot dataset for models MLP, LSTM, CNN presented in the main text?
>
> **A6:** Cross-architecture performance is an important metric reported in all dataset condensation papers. MLP, CNN and LSTM perform well on time series forecasting[1] and are widely used in the evaluation of dataset condensation[2]. Thus we present the result in the main text.
>
> [1] Are Transformers Effective for Time Series Forecasting?
>
> [2] Dataset Condensation for Time Series Classification via Dual Domain Matching
>
> ---
>
> **Q2:** Add a nonlinear expert?
>
> **A7:** Please refer to App.A1 and Eq(16). Given nonlinear models, our method ignores the 2nd and higher order terms in Taylor Expansion. We conduct experiments to show that ignorance is acceptable. We use nonlinear models CNN, 3-layer MLP(ReLU activated) as expert models. Average MAE of 5 models is reported below. The results show that CondTSF performs well with nonlinear models.
>
> - CNN
>
> ||Ex.Rate|Wea.|Elec.|Tra.|ETTm1|ETTm2|ETTh1|ETTh2|
> |-|-|-|-|-|-|-|-|-|
> |MTT|0.372|0.314|0.482|0.662|0.550|0.347|0.644|0.371|
> |MTT+CondTSF|0.140|0.246|0.357|0.451|0.482|0.237|0.460|0.297|
> |TESLA|0.378|0.310|0.516|0.655|0.544|0.359|0.634|0.365|
> |TESLA+CondTSF|0.134|0.253|0.374|0.528|0.499|0.251|0.473|0.293|
> |DATM|0.331|0.335|0.504|0.587|0.566|0.318|0.633|0.349|
> |DATM+CondTSF|0.137|0.291|0.355|0.452|0.518|0.231|0.451|0.290|
>
> - 3-layer MLP(ReLU activated)
>
> ||Ex.Rate|Wea.|Elec.|Tra.|ETTm1|ETTm2|ETTh1|ETTh2|
> |-|-|-|-|-|-|-|-|-|
> |MTT|0.364|0.311|0.475|0.633|0.564|0.362|0.615|0.354|
> |MTT+CondTSF|0.139|0.248|0.375|0.501|0.493|0.245|0.423|0.285|
> |TESLA|0.352|0.297|0.525|0.594|0.541|0.341|0.606|0.337|
> |TESLA+CondTSF|0.128|0.252|0.397|0.488|0.487|0.250|0.438|0.283|
> |DATM|0.326|0.349|0.517|0.622|0.558|0.329|0.593|0.350|
> |DATM+CondTSF|0.141|0.254|0.385|0.496|0.498|0.234|0.437|0.286|
>
> ---
>
> **Q3:** What's the novelty beyond time series?
>
> **A8:**
> - All previous dataset condensation methods based on classification perform poorly on regression. The core novelty of our method is that it is the first dataset condensation method focusing on regression and significantly enhances the performance of all backbone methods.
> - In time series forecasting, when generating training data, the data is usually sampled overlap from the dataset. For instance, the data points can be sampled as $x_{0:24}, x_{4:28}$, etc. Please refer to Eq(12), inspired by the overlap property, we utilize an additive method to gradually update the synthetic data instead of directly using $M_f(s_{t':t'+m})$ to overwrite $s_{t'+m:t'+m+n}$ to avoid vibrations.
> - Additionally, other content related to time series can be explored in future works. We presented a simple yet effective method, which can serve as a prototype.

---

> > ### Comment · Reviewer_TXkn · 2024-08-12
> >
> > Thank you for your rebuttal. I was expecting improvements on the writing quality of the paper. Would it be possible to cite in comment a few rewrites if any?

---

> > > ### Author Response · Authors · 2024-08-12
> > > **Plan of rewriting part of our paper**
> > >
> > > Dear reviewer TXkn,
> > >
> > > Thank you for your comment and advice on improving the writing of our paper. We apologize for the ambiguity that exists in the current version. We will rewrite the following part of our paper.
> > >
> > > 1. We will clarify the source of the train set $f$, test set $x$, and synthetic dataset $s$ in Sec.3.
> > > 2. We will rewrite App.A1 and texts relate to Eq(5) to clarify the test error and the meaning of $t'$. We will also add an explanation of the effects when nonlinear models are used as expert models.
> > > 3. We will add a detailed introduction of MTT on how to optimize Eq(7) and Eq(8) in the appendix to clarify the core novelty of our paper.
> > > 4. We will add the complete results of using nonlinear models as expert models in the appendix to show that our method also works well with nonlinear experts.
> > > 5. We will add an explanation in Sec4.4 of how the additive method relates to the features of time series.
> > >
> > > Meanwhile, we are eager for the opportunity to address any remaining concerns you might have before the discussion period ends. We sincerely appreciate your and other reviewers' dedication and time invested in reviewing our work.
> > >
> > > Best,
> > >
> > > Authors

---

### Official Review · Reviewer_ivQc · 2024-07-14

**Soundness:** 3
**Presentation:** 3
**Contribution:** 3
**Rating:** 7
**Confidence:** 3

**Summary:**

This paper provides a simple fix on the parameter-matching based dataset condensation methods for time series forecasting tasks. Besides matching the parameters of the model $M_f$ trained on the full training set and $M_s$ trained on the smaller synthetic dataset, it additionally induces the $M_f$ to perform well in the synthetic data too. It theoretically shows that the additional regularization contributes to an surrogate upper bound of the original objective, and empirically proves that the fix help improve various dataset condensation methods.

**Strengths:**

1. The paper is well-written with clear structure and proper organization, and the motivation of the proposed condTSF plugin is solidly grounded by theoretical analyses
2. The proposed method is simple and easy to implement.
3. Impressive improvement are observed in the experiments on various baselines and benchmarks.

**Weaknesses:**

A few details are not clear. See questions.

**Questions:**

1. In line 162, the author claims that it is "observed" that models initialized with different samples of parameter values predict similarly after trained on the full dataset given arbitrary input. Is it supported by any evidences, either theoretically or empirically?
2. Some clarity on optimizing gradient term can be provided. For example, when the parameters are matched and $M_s$ are well fitted, why the label error of $M_f$ on $s$ will increase?

**Limitations:**

The limitation that the analysis only applies to linear models is discussed in the paper.

---

> ### Author Rebuttal · Authors · 2024-08-06
>
> **Q1:**
> Is the claim "models initialized with different samples of parameter values predict similarly after trained on the full dataset given arbitrary input" in line 162 or Eq(9) supported by any evidence?
>
> **A1:**
> We apologize for the ambiguity. Please refer to Figure 3 and line 166-168, we support this empirical observation with experiments. The large orange points in the figure are input data, and the yellow and blue points are predictions of the models initiated with different parameters. We used MDS algorithm to reduce the dimension of the original prediction and data. MDS algorithm preserves the distance between points, which means points closer to each other in high dimensions are also close in low dimensions. It can be observed that the predictions of the models are similar despite their initial parameters are different.
>
> ----
>
> **Q2:**
> Why the label error on $s$ will increase when the parameters of $M_s$ are matched.
>
> **A2:** We apologize for the ambiguity.
> - Firstly, The label error is used to evaluate the synthetic dataset $s$ instead of the expert model $M_f$. Please refer to Thm.2 and Eq(11), label error is an upper bound of the **value term**. Therefore label error ensures the similarity between prediction values of $M_f$ and $M_s$.
> - Secondly, the parameter loss is an upper bound related to the **gradient term**. It only ensures $M_s$ and $M_f$ have similar gradients on the input data, yet it does not ensure their value of predictions are similar.
> - To sum up, our analysis in Sec.4.1 and Thm.1 showed that **value term** and **gradient term** are decoupled in the optimization. Parameter loss relates to the upper bound of **gradient term** while label error is the upper bound of **value term**. Therefore, even when parameter loss is small, the label error can still be large.

---

### Author Rebuttal · Authors · 2024-08-06

Dear reviewers and area chairs,

We thank all reviewers and area chairs for their valuable time and comments.

We have responded to each reviewer individually to address any comments. We would like to give a brief summary.

1. We clarify some of the meanings of the notations in our paper to address the concern.
2. We provide information on backbone methods and explain the novelty of the work. We also explain that our proposed method has designs that relate to the features of time series.
3. We add experiments using nonlinear models as expert models and show that CondTSF also works well with nonlinear expert models.
4. We explain the practical applications of dataset condensation and show that dataset condensation for time series forecasting has pragmatic values.
5. We show that deriving the upper bound of the **gradient term** is necessary because there exists unavailable test data in the original form.
6. We show that **dataset condensation** and **model distillation** are distinct tasks, and thus the methods are not comparable.
7. We add experiments using a low pass filter to perform target smoothing on synthetic data and show that CondTSF is significantly more effective than target smoothing.
8. We add experiments using only CondTSF(optimizing only **value term**) and show the impacts of **gradient term** and **value term**.

Again, we thank all reviewers and area chairs!

Best,

Authors

---

### Decision · Program_Chairs · 2024-09-25

**Decision:**

Accept (poster)

**Comment:**

This paper introduces a simple and effective method for condensing time series datasets, which can be implemented as a one-line plugin for time series forecasting models corresponding to an additional regularization term.
The paper develops a theory showing that this regularization term allows for upper bounding the objective of the dataset condensation problem.
The resulting algorithm is then empirically validated on 8 time series datasets which confirm the effectiveness of the proposed method.
The reviewers generally agree that the paper is clearly organized and structured. Other strengths of the paper include the simplicity of the resulting method, the theoretical analysis, and the strong performance in experiments. However, some reviewers raised concerns about the ambiguous notation, the lack of motivation and significance of addressing dataset condensation for time-series, and the limited evaluation.
The rebuttals satisfactorily addressed most of these concerns, including adding additional experiments to verify that the methods is applicable to nonlinear models, and including comparisons against more baselines like target smoothing on synthetic data.
Overall, the reviewers found the paper to be a solid enough contribution to warrant acceptance. Reviewers however recommend that acceptance be conditional on the additional results in the rebuttals and notation and presentation improvements be included and prominently showcased in the camera-ready version of the paper.